# DINGO: Constrained Inference for Diffusion LLMs

**Tarun Suresh**\*, **Debangshu Banerjee**,\* **Shubham Ugare, Sasa Misailovic, Gagandeep Singh**
Department of Computer Science,
University of Illinois Urbana-Champaign

## Abstract

Diffusion LLMs have emerged as a promising alternative to conventional autoregressive LLMs, offering substantial potential for improving runtime efficiency. However, existing diffusion models fail to provably enforce user-specified formal constraints, such as regular expressions, which makes them unreliable for tasks that require structured outputs, such as fixed-schema JSON generation. Unlike autoregressive models, which generate tokens sequentially, diffusion LLMs predict a block of tokens in parallel. This parallelism makes traditional constrained decoding algorithms, designed to enforce constraints with sequential token prediction, ineffective at preserving the true output distribution. To address this limitation, we propose DINGO, a dynamic programming-based constrained decoding strategy that is both efficient and provably distribution-preserving. DINGO enables sampling of output strings with the highest probability under the model's predicted distribution while strictly adhering to any user-specified regular expression. On standard symbolic math and JSON generation benchmarks, DINGO achieves up to a $68\%$ points of improvement over unconstrained inference. The code is available at `DINGO`.

## 1 Introduction

Autoregressive LLMs demonstrate impressive performance across a wide range of tasks, including logical reasoning [Pan et al., 2023], theorem proving [Yang et al., 2023], and code generation [et. al., 2021]. However, because they generate one token at a time, they can be slow when producing long responses. Recent work has explored using diffusion models to accelerate token generation by predicting blocks of tokens in parallel. For tasks such as logical reasoning, where the LLM output is fed into symbolic solvers like Z3 [Fedoseev et al., 2024], syntactic correctness of the output is essential. Prior works [Poesia et al., 2022, Ugare et al., 2024a, Loula et al., 2025] have shown that LLMs frequently make syntactic and semantic errors, often generating structurally invalid outputs that cause downstream tasks to fail due to unparsable input. To mitigate this issue, constrained decoding has emerged as a promising approach that provably ensures structural correctness by projecting the LLM output onto a set of valid strings, typically defined by a regular grammar or, more generally, a context-free grammar (CFG). However, existing constrained decoding techniques are designed specifically for autoregressive LLMs and rely on their step-by-step generation process to prune invalid tokens that cannot lead to structurally valid outputs. At each generation step, the decoder selects the highest-probability token from the set of valid options, based on the LLM's output distribution.

In contrast, diffusion LLMs predict blocks of tokens in parallel without sequential dependencies, making existing constrained decoding algorithms incompatible. Furthermore, greedy token selection in autoregressive models maximizes the probability locally at each step but can be suboptimal over an entire sequence, potentially leading to structurally valid yet lower-quality outputs that fail to maximize the overall probability of valid strings. Lew et al. [2023], Park et al. [2024b] have reported this distortion in output distribution for autoregressive LLMs under constrained decoding. Therefore,

---

\*Equal contributing authors ordered randomly

any constrained decoding algorithm for diffusion LLMs should also ensure that enforcing formal constraints does not come at the cost of distorting the true output distribution.

**Key Challenges:** Diffusion LLMs generate a block of tokens starting from a fully masked string composed of special mask tokens $\perp$, and iteratively unmask one or more tokens at each step until producing a fully unmasked output. Each unmasking step (referred to as a diffusion step) can unmask tokens at arbitrary positions in the block, with no left-to-right sequential dependency across steps. As a result, designing constrained decoding for diffusion LLMs requires addressing the following:

- **RQ1:** Efficiently detecting invalid tokens and restricting token choices at each diffusion step to ensure the final unmasked string is always structurally correct.
- **RQ2:** Ensuring the generated token block maximizes the probability under the output distribution.

**Contributions:** We present the first constrained decoding algorithm for diffusion LLMs, making the following contributions:

- We introduce DINGO, the first constrained decoding algorithm for diffusion LLMs that supports any user-specified regular expression. DINGO provably ensures that the output string is always a valid prefix of some string in the target regular language.
- DINGO uses dynamic programming to ensure that the output string achieves the maximum probability among all valid strings over the output block with respect to the true output distribution. This approach guarantees scalability while maintaining optimality (e.g., maximizing the probability), in contrast to existing methods such as [Park et al., 2024b], which rely on repeated resampling. Resampling-based methods are computationally expensive and unsuitable for practical deployment.
- Extensive experiments on multiple open-source diffusion LLMs and benchmarks show that DINGO significantly outperforms standard unconstrained decoding, achieving up to a $68\%$ improvement on challenging tasks such as the GSM-symbolic benchmark for symbolic reasoning [Mirzadeh et al., 2024] and a JSON generation benchmark [NousResearch, 2024].

**Roadmap:** We provide the necessary background in Section 2, formalize constrained decoding for diffusion LLMs in Section 3, describe the DINGO algorithm along with its correctness and optimality proofs in Section 4, and present experimental results in Section 5.

## 2   Background

**Notation:** : In the rest of the paper, we use small case letters $x$ for constants, bold small case letters ($\boldsymbol{x}$) for strings, capital letters $X$ for functions, $\cdot$ for string concatenation, $|\boldsymbol{x}|$ to denote the length of $\boldsymbol{x}$.

**Diffusion LLM:** The diffusion LLM $\mathcal{L}_{m,n} : V^m \to V^n$ processes finite strings $\boldsymbol{p} \in V^m$ over a finite alphabet $V$ including the special mask symbol $\perp$ and produces the output string $\boldsymbol{o} \in V^n$. Typically $\boldsymbol{o} = \boldsymbol{p} \cdot \boldsymbol{r}$ with length $n$ represents the entire output string of $\mathcal{L}$ where $\boldsymbol{p}$ is the input prompt, $\boldsymbol{r}$ is the response, and $m + |\boldsymbol{r}| = n$. $\mathcal{L}$ can compute the response $\boldsymbol{r}$ over a single block Austin et al. [2021], Ye et al. [2025], Nie et al. [2025] in pure diffusion setup or over multiple blocks i.e. $\boldsymbol{r_1} \cdot \boldsymbol{r_2} \cdots \boldsymbol{r_k}$ in a semi-autoregressive setup where different blocks are computed sequentially from left to right [Han et al., 2023, Arriola et al., 2025].

At a high level, to compute a block of tokens of size $d$, $\mathcal{L}$ pads the prompt $\boldsymbol{p}$ with a fully masked suffix, resulting in $\boldsymbol{p} \cdot \perp^d$, where $\perp^d$ denotes a sequence of $d$ special mask tokens $\perp$. The model then iteratively unmasks a subset of these tokens at each step, ultimately producing a fully unmasked output string $\boldsymbol{o}$. Each such step is referred to as a diffusion step, and $\mathcal{L}$ typically applies $T$ diffusion steps to compute $\boldsymbol{o}$. The number of steps $T$ is usually a fixed, predetermined constant satisfying $T < d$, which enables greater scalability compared to their autoregressive counterparts.

**Definition 2.1** (Diffusion step). A diffusion step $f_n : V^n \times \mathbb{N} \to V^n$ applies a single unmasking step to a masked (or, a partially masked) string of length to compute a new masked (or, possibly unmasked) string of the same length. The first argument represents the input string appended with the output block while the second argument dictates the number of masked tokens in the output string.

Each diffusion step $f_n$ consists of two components: a transformer step $\mathcal{N}_n : V^n \to \mathbb{R}_+^{|V| \times n}$, which predicts the token probability distribution at each output position, and a mask prediction step

$\mathcal{M}_n : \mathbb{R}_+^{|V| \times n} \times \mathbb{N} \to \mathbb{R}_+^{|V| \times n}$, which determines which token positions to remask. Typically, for each position, the mask prediction step identifies the token with the highest probability and compares these maximum probabilities across positions. $\mathcal{M}_n$ then greedily remasks positions with relatively lower max-probability scores [Nie et al., 2025] and produces the modified token distribution. Further details about $\mathcal{N}_n$ and $\mathcal{M}_n$ are in Appendix A.

Formally, the diffusion step is defined as $f_n(\boldsymbol{x}_{i-1}, i) = D_{m,n}(\mathcal{M}_n(\mathcal{N}_n(\boldsymbol{x}_{i-1}), i))$ where $D_{m,n} : \mathbb{R}_+^{|V| \times n} \to V^n$ is the decoder. We now use the diffusion step to formally define the diffusion LLM for generating strings of length $n$ in either a single-block or multi-block setting.

**Definition 2.2** (Single block diffusion LLM). A diffusion LLM that outputs a block of $d$ tokens given an input $\boldsymbol{p} \in V^m$ using $T$ diffusion steps is a function $\mathcal{L}_{m,n} : V^m \to V^n$, where $n = m + d$, and the output is $\boldsymbol{o} = \boldsymbol{p} \cdot \boldsymbol{r} = \mathcal{L}_{m,n}(\boldsymbol{p})$. Let $f_n : V^n \times \mathbb{N} \to V^n$ denote a single diffusion step, and let $P_{m,n} : V^m \to V^n$ be the padding function. Then the output is computed as $\boldsymbol{o} = \mathcal{L}_{m,n}(\boldsymbol{p}) = \boldsymbol{x}_T$, where: $\boldsymbol{x}_0 = P_{m,n}(\boldsymbol{p}) = \boldsymbol{p} \cdot \perp^d$ and $\boldsymbol{x}_i = f_n(\boldsymbol{x}_{i-1}, i)$ for $1 \leq i \leq T$.

**Definition 2.3** (Semi Autoregressive diffusion LLM). In the semi-autoregressive setup, given an input $\boldsymbol{p} \in V^m$, the output $\boldsymbol{o} \in V^{m+d \times k}$ is generated over $k$ blocks, where each block is computed via a call to the single block diffusion model. The output of the $i$-th diffusion model call is $\boldsymbol{x}_i = \mathcal{L}_{m_i, n_i}(\boldsymbol{x}_{i-1})$, with $\boldsymbol{x}_0 = \boldsymbol{p}$ and the final output $\boldsymbol{o} = \boldsymbol{x}_k$. The input and output lengths for each block are defined as $m_i = m + (i-1) \times d$ and $n_i = m + i \times d$ for all $1 \leq i \leq k$.

**DFA and regular expression:** We provide necessary definitions regarding regular expression.

**Definition 2.4.** (DFA) A DFA $D_{\mathcal{R}} = (Q, \Sigma, \delta, q_0, F)$ for a regular expression $\mathcal{R}$ is a finite-state machine that deterministically processes input strings to decide membership in the language $L(\mathcal{R}) \subseteq \Sigma^*$ defined by $\mathcal{R}$. It consists of states $Q$, a start state $q_0$, a set of accepting states $F$, and transition rules $\delta : Q \times \Sigma \to Q$ and the input alphabet $\Sigma$.

**Definition 2.5** (extended transition function). The extended transition function $\delta^* : \Sigma^* \times Q \to Q$ maps an input $(\boldsymbol{w}, q)$ to the resulting state $q_r$, obtained by sequentially applying $\delta$ to each character $c_i$ in $\boldsymbol{w} = c_1 \cdots c_{|\boldsymbol{w}|}$, starting from state $q$.

**Definition 2.6** (Live DFA states). Given a DFA $(Q, \Sigma, \delta, q_0, F)$, let $Q_l$ represent the set of live states such that $q \in Q_l$ iff $\exists w \in \Sigma^*$ s.t. $\delta^*(\boldsymbol{w}, q) \in F$.

# 3 Optimal Constrained Decoding

We formalize the correctness and optimality of constrained decoding for any diffusion LLM with respect to a user-defined regular expression $\mathcal{R}$. Given $\mathcal{R}$, let $L(\mathcal{R}) \subseteq \Sigma^* \subseteq (V \setminus \perp)^*$ denote the set of all finite strings that satisfy the expression $\mathcal{R}$.

**Correctness:** A valid constrained decoding algorithm must ensure that the output string always remains a valid *prefix* of some string in $L(\mathcal{R})$, effectively eliminating any output that cannot be extended into valid completions. By treating the output string as a prefix rather than a fully completed string, we can accommodate the semi-autoregressive setup, where blocks of tokens are appended to the right of the current output. This approach avoids prematurely rejecting strings that may lead to valid completions in subsequent blocks and also aligns with the notion of correctness adopted in existing constrained decoding algorithms for the autoregressive LLM [Ugare et al., 2024b, Banerjee et al., 2025a]. We denote the set of all valid prefixes of $L(\mathcal{R})$ as $L_P(\mathcal{R})$.

Each diffusion step $f_n$ produces a string over the vocabulary $V$, which may include one or more special mask tokens $\perp$. These tokens act as placeholders for actual (non-mask) tokens that will be filled in during future diffusion steps. To account for these future substitutions, we define a masked (or partially masked) string as valid if there exists a replacement for all mask tokens such that the resulting fully unmasked string is a valid prefix of some string in $L(\mathcal{R})$. To formalize this notion, we first define the *substitution set*, which represents the set of fully unmasked strings obtained by replacing all mask tokens in a masked or partially masked string. We then use substitution sets to define the correctness of the constrained decoder.

**Definition 3.1** (Substitution Set). Given a masked (or, partially masked) string $\boldsymbol{x} \in V^n$, the *substitution set* $\mathcal{S}(\boldsymbol{x}) \subseteq (V \setminus \{\perp\})^n$ is the set of all fully unmasked strings obtained by replacing each occurrence of $\perp$ in $\boldsymbol{x}$ with a token from $V \setminus \{\perp\}$. For unmasked strings with no $\perp$, $\mathcal{S}(\boldsymbol{x}) = \{\boldsymbol{x}\}$

**Definition 3.2** (Correctness of Constrained decoder). Any deterministic decoder $D_{m,n,\mathcal{R}}$ : $\mathbb{R}_+^{|V| \times n} \to V^n$ is a valid constrained decoder if, for all $n \in \mathbb{N}$, input prompt $\boldsymbol{p}$ and for any output distribution $\mathcal{D}_n$ provided as $n$ probability vectors each of size $|V|$, there exists an unmasked string $\boldsymbol{x}$ in the substitution set $\mathcal{S}(D_{m,n,\mathcal{R}}(\mathcal{D}_n))$ of the decoded output such that actual response $\boldsymbol{p} \cdot \boldsymbol{r} = \boldsymbol{x}$ is a valid prefix i.e., $\boldsymbol{r} \in L_P(\mathcal{R})$. [2]

**Optimality:** Given a distribution $\mathcal{D}_n$ and a regular expression $\mathcal{R}$, the set of decodings that are valid prefixes for $\mathcal{R}$ (as defined in Definition 3.2) may not be unique. An optimal constrained decoder selects, among all valid strings, the string that maximizes the probability under $\mathcal{D}_n$. The output distribution $\mathcal{D}_n$ is represented as $n$ vectors $\boldsymbol{v}_1, \ldots, \boldsymbol{v}_n$, each of size $|V|$, where the $i$-th vector $\boldsymbol{v}_i$ captures the token distribution at position $i$. For any masked position $j$, $\boldsymbol{v}_j$ assigns probability 1 to the mask token $\perp$ and 0 to all other tokens. Assuming the input prompt has length $m$, the token distribution of the actual response is given by $\boldsymbol{v}_{m+1}, \ldots, \boldsymbol{v}_n$. For any output string $\boldsymbol{r} = t_{m+1} \ldots t_n$, let $P(\boldsymbol{r} \mid \boldsymbol{v}_{m+1} \ldots \boldsymbol{v}_n)$ denote the probability of the string $\boldsymbol{r}$ under the output distribution. Then, the optimal constrained decoding can be formalized as follows:

$$\boldsymbol{r}^* = \arg\max_{\boldsymbol{r}} P(\boldsymbol{r} \mid \boldsymbol{v}_{m+1} \ldots \boldsymbol{v}_n) \text{ s.t. } \exists \boldsymbol{x} \in V^*.(\boldsymbol{x} \in \mathcal{S}(\boldsymbol{r})) \wedge (\boldsymbol{x} \in L_P(\mathcal{R})) \quad (1)$$

Since the token distributions $\boldsymbol{v}_{m+1}, \ldots, \boldsymbol{v}_n$ are independent across positions, the probability of the string $\boldsymbol{r}$ can be written as $P(\boldsymbol{r} \mid \boldsymbol{v}_{m+1} \ldots \boldsymbol{v}_n) = \prod_{i=m+1}^{n} \boldsymbol{v}_i[t_i]$ where $\boldsymbol{v}_i[t_i]$ denotes the probability assigned to token $t_i$ by the vector $\boldsymbol{v}_i$. Using this, we can rewrite the optimization problem from Eq. 1 as follows:

$$\boldsymbol{r}^* = \arg\max_{\boldsymbol{r}=t_{m+1} \cdots t_n} \prod_{i=m+1}^{n} \boldsymbol{v}_i[t_i] \text{ s.t. } \exists \boldsymbol{x} \in V^*.(\boldsymbol{x} \in \mathcal{S}(\boldsymbol{r})) \wedge (\boldsymbol{x} \in L_P(\mathcal{R})) \quad (2)$$

## 4 DINGO Algorithm

The search space for Eq. 2 is exponential– $|V|^d$, where $d = n - m$ denotes the block length, making naive enumeration-based methods impractical. To efficiently retrieve the optimal output string $\boldsymbol{r}^*$ from Eq. 2, DINGO leverages dynamic programming. Given a regular expression $\mathcal{R}$, it first modifies the transition function to handle the mask symbol $\perp$, which is then utilized during inference.

### 4.1 Precomputation

For a user-provided $\mathcal{R}$ and the corresponding DFA $D_{\mathcal{R}} = (Q, \Sigma, \delta_{\mathcal{R}}, q_0, F)$ (referred to as character-level DFA) with $\Sigma \subseteq (V \setminus \perp)$, we first construct the token-level DFA $D_t = (Q, (V \setminus \perp), \delta_t, q_0, F)$ recognizing $L(\mathcal{R})$ over strings generated by $\mathcal{L}$. A single token $\boldsymbol{t} \in (V \setminus \perp)$ can span across multiple characters in $\Sigma$ i.e. $\boldsymbol{t} = c_1 \cdots c_l$ where $c_i \in \Sigma$. To construct the token-level transition function $\delta_t : Q \times (V \setminus \perp) \to Q$, we process each token $\boldsymbol{t} \in (V \setminus \perp)$ and state $q \in Q$ by executing the character-level DFA $D_{\mathcal{R}}$ on the sequence of constituent characters $c_1 \cdots c_l$, starting from state $q$, and record the resulting state $q_r$. We then define the token-level transition as $\delta_t(q, \boldsymbol{t}) = q_r$.

To handle the special mask token $\perp \in V$, we define the transition function $\delta_\perp : Q \to 2^Q$. For each state $q \in Q$, $\delta_\perp(q)$ returns the set of states $Q_r \subseteq Q$ that are reachable via a single token transition using $\delta_t$. Formally, $\delta_\perp(q) = \{q' \mid q' = \delta_t(q, \boldsymbol{t}); \boldsymbol{t} \in (V \setminus \perp)\}$. Since $\delta_\perp$ may return multiple states, it resembles the transition function of a non-deterministic finite automaton (NFA). The precomputation step combines $\delta_t$ and $\delta_\perp$ to define $\delta : Q \times V \to 2^Q$, which is used in the dynamic programming step. Using the token-level DFA $D_t$, we also construct the set of live states $Q_l \subseteq Q$ (Definition 2.6).

$$\delta(q, \boldsymbol{t}) = \begin{cases} \{\delta_t(q, t)\} & \text{if } t \in (V \setminus \perp), \\ \delta_\perp(q) & \text{if } t = \perp. \end{cases}$$

### 4.2 DINGO Dynamic Programming

Before going into details, we present two key observations that lead to the decoding algorithm.

---

[2]More precisely, if there exists at least one $\boldsymbol{r}$ that is a valid prefix (i.e., $\boldsymbol{r} \in L_P(\mathcal{R})$), then constrained decoding is always capable of retrieving one of them.

**Observation 1:** Determining whether a fully unmasked string $r = t_1 \cdots t_d \in (V \setminus \perp)^*$ is a valid prefix is equivalent to checking whether the resulting state $q_r$, obtained by applying $\delta$ to the sequence $t_1 \cdots t_d$ starting from $q_0$, is live. Similarly, for a partially (or fully) masked string $r_\perp$, applying $\delta$ to $t_1 \cdots t_d$ yields a set of resulting states $Q_r$. In this case, $r_\perp$ is a valid prefix if and only if any state $q \in Q_r$ is live (Definition 3.2).

**Observation 2:** For optimality, it is sufficient to track the maximum probability path from the start state $q_0$ to each resulting state in $Q_r$. Once these paths are computed, we select the one with the highest probability that leads to a live state. The corresponding string is the optimal string $r^*$ (or one of the optimal strings in case of multiple optimal solutions) for the optimization problem in Eq. 2.

Based on these observations, the main challenge is to efficiently maintain the maximum probability path to each reachable state in $Q_r$. We address this using a dynamic programming (DP) approach, similar to traditional graph-based DP algorithms such as [Forney, 1973].

**DP states:** For each token position $1 \leq i \leq d$ in the block, the DP maintains: a) $W[i, q]$, which records the maximum probability with which a state $q \in Q$ can be reached from the start state $q_0$ via transitions on some token sequence with length $i$; and b) $Pr[q, i]$, which stores the last transition, i.e., the previous state and the corresponding token, that led to the maximum probability stored in $W[i, q]$. If a state $q$ is unreachable, then $W[i, q] = 0$. Formally, given the probability vectors $\boldsymbol{v}_1, \ldots, \boldsymbol{v}_i$, $W[i, q]$ is defined as follows where $\delta_t^*$ is extended transition function (Definition 2.5).

$$W[i, q] = \max_{t_1 \ldots t_i} \prod_{j=1}^{i} \boldsymbol{v}_j[t_j] \quad \text{s.t. } q = \delta_t^*(t_{m+1} \cdots t_n, q_0)$$

**DP state update:** Given the states at token position $i$, we describe the computation for position $i + 1$. Initially, $W[i, q] = 0$ for all $q \neq q_0$, and $W[i, q_0] = 1$ (lines $1 - 3$ in Algo. 1). To compute $W[i + 1, q]$ for each $q \in Q$, we consider all tokens $t \in V$ (including the mask token $\perp$) that can transition to $q$ from some previous state $q'$ at step $i$. Among all such transitions, we select the one with the highest probability and add it to the maximum probability path reaching $q'$ at step $i$. The value $Pr[i + 1, q]$ stores the previous state and token that lead to the maximum probability path to $q$ at step $i + 1$ (lines $12 - 15$ in Algo. 1). Formally,

$$V_{i+1}(q, q') = \begin{cases} \max_{t \in V} \boldsymbol{v}_{i+1}(t) \text{ s.t. } q \in \delta(q', t) \\ 0 \text{ if } q, q' \text{ are not connected} \end{cases} \quad W[i + 1, q] = \max_{q' \in Q} W[i, q'] \times V_{i+1}(q, q')$$

**Path construction:** We consider all reachable states $q$ at the end of the block with $W[d, q] > 0$. Among the live states $q_l \in Q_l$ satisfying this condition, we select the state $q_{\max}$ with the highest value of $W[d, q_l]$. We then use $Pr$ to iteratively reconstruct the token sequence backward that forms the maximum probability path starting from $q_{\max}$ and ending at $q_0$ (lines $20 - 22$ in Algo. 1).

**Semi-autoregressive setup:** In semi-autoregressive setup, we may not start from DFA start state $q_0$ since one or more blocks of tokens $r_1 \cdots r_l$ may have been generated in left the current block. Provide the string $r_1 \cdots r_l$ ends at a live state $q_l$, we can apply dynamic programming approach with the intializtion $W[0, q_l] = 1$ and $W[0, q] = 0$ for all state $q \neq q_l$. Details are in Appendix D.

### 4.3 Correctness of DINGO

**Proposition 4.1.** *[Correctness] Given any regular expression $\mathcal{R}$, input prompt $\boldsymbol{p} \in V^m$, block length $d$, output distribution $\mathcal{D}_{m+d} = \boldsymbol{v}_1 \ldots \boldsymbol{v}_{m+d}$, if $L_P(\mathcal{R}) \cap (V \setminus \perp)^d \neq \{\}$ and $r \sim \boldsymbol{v}_{m+1} \ldots \boldsymbol{v}_{m+d}$ be the decoded string, then $\exists \boldsymbol{x} \in V^*. (\boldsymbol{x} \in \mathcal{S}(\boldsymbol{r})) \wedge (\boldsymbol{x} \in L_P(\mathcal{R}))$ holds.*

**Proof sketch:** DINGO ensures that if a state $q \in Q$ is reachable in $i$ tokens, then $W[i, q] > 0$ for all $1 \leq i \leq d$. Since $L_P(\mathcal{R}) \cap (V \setminus \perp)^d \neq \{\}$, there exists a state $q_l \in Q_l$ that is reachable in $d$ steps. Therefore, $W[d, q_{max}] > 0$ (see line 16 in Alg.1). Consequently, there exists a sequence $\boldsymbol{x} \in \mathcal{S}(\boldsymbol{r})$ such that $\delta^*(\boldsymbol{x}, q_0) = q_{max} \in Q_l$, implying that $\boldsymbol{x} \in L_P(\mathcal{R})$. Formal proof is in AppendixB.

**Proposition 4.2.** *[Optimality] Given any regular expression $\mathcal{R}$, input prompt $\boldsymbol{p} \in V^m$, block length $d$, output distribution $\mathcal{D}_{m+d} = \boldsymbol{v}_1 \ldots \boldsymbol{v}_{m+d}$, if $L_P(\mathcal{R}) \cap (V \setminus \perp)^d \neq \{\}$ and $r^* \sim \boldsymbol{v}_{m+1} \ldots \boldsymbol{v}_{m+d}$ be the decoded string, then for any valid string $r'$ satisfying $\exists \boldsymbol{x} \in V^*. (\boldsymbol{x} \in \mathcal{S}(\boldsymbol{r}')) \wedge (\boldsymbol{x} \in L_P(\mathcal{R}))$, $P(\boldsymbol{r}' \mid \boldsymbol{v}_{m+1} \ldots \boldsymbol{v}_n) \leq P(\boldsymbol{r}^* \mid \boldsymbol{v}_{m+1} \ldots \boldsymbol{v}_n)$.*

**Proof Sketch:** Formal proof is in Appendix B.

---

**Algorithm 1** DINGO DP

---

**Require:** $q_0$, block length $d$, probability vectors $\boldsymbol{v}_1, \ldots \boldsymbol{v}_d$ for the current block, $Q_l$, $Q$, $\delta$.

1:  $W[0, q] \leftarrow 0$ for all $(q \in Q) \wedge (q \neq q_0)$
2:  $W[0, q_0] \leftarrow 1$
3:  $Pr[0, q] \leftarrow (\text{None}, \text{None})$ for all $(q \in Q)$         $\triangleright$ Initialization of the DP
4:  $V_i \leftarrow \{\}$ for all $i \in \{1, \ldots, d\}$    $\triangleright$ maximum token probability transtion $(q' \rightarrow q)$ at position $i$
5:  $T_i \leftarrow \{\}$ for all $i \in \{1, \ldots, d\}$     $\triangleright$ token for the maximum probability transition $(q' \rightarrow q)$
6:  **for** $i \in \{1, \ldots, d\}$ **do**
7:   **for** $(q \in Q)$ **do**
8:    **for** $t \in V$ **do**
9:     $q' \leftarrow \delta(q, t)$
10:     $V_i(q, q'), T_i(q, q') \leftarrow \text{MaxTransition}(\boldsymbol{v}_i, t, q, q')$
11: **for** $i \in \{1, \ldots, d\}$ **do**             $\triangleright$ DP computation loop
12:   **for** $(q \in Q) \wedge (q' \in Q)$ **do**
13:    **if** $W[i, q] < W[i-1, q'] \times V_i(q, q')$ **then**
14:     $W[i, q] \leftarrow W[i-1, q'] \times V_i(q, q')$    $\triangleright$ Update maximum probability path to $q$
15:     $Pr[i, q] \leftarrow (q', T_i(q, q'))$       $\triangleright$ Update the parents accordingly
16: $q_{max} \leftarrow \arg\max_{q \in Q_l} W[d, q]$
17: **if** $W[d, q_{max}] = 0$ **then**           $\triangleright$ No valid prefixes
18:   **return** None, $q_{max}$
19: $\boldsymbol{r}^* \leftarrow \{\}, q_{curr} \leftarrow q_{max}$
20: **for** $i \in \{d, \ldots, 1\}$ **do**         $\triangleright$ Decoding the optimal string $\boldsymbol{r}^*$
21:   $q_{curr}, t \leftarrow Pr[i, q_{curr}]$
22:   $\boldsymbol{r}^* \leftarrow \boldsymbol{r}^* \cdot t$
23: **return** reverse($\boldsymbol{r}^*$), $q_{max}$

---

## 4.4   DINGO algorithm

Algorithm 1 presents DINGO steps. The two main loops dominating its computational complexity involve calculating transition costs and performing the DP updates respectively.

First, for each of the $d$ time steps, the algorithm computes the optimal single-token transition costs $V_i(q_s, q_t)$ between all source states $q_s \in Q$ and target states $q_t \in Q$. This is achieved by iterating through each source state $q_s$, each token $t \in V$, and then for each state $q_t$ reached from $q_s$ via $t$ (i.e., $q_t \in \delta(q_s, t)$), updating the cost $V_i(q_s, q_t)$ with $\boldsymbol{v}_i[t]$ if it is better. The complexity for this part is $O(d \cdot (|Q|^2 + \sum_{q_s \in Q} \sum_{t \in V} |\delta(q_s, t)|))$. The sum $\sum_{q_s} \sum_t |\delta(q_s, t)|$ represents the total number of transitions, $N_{\text{trans}} = O(|Q| \cdot |V| + |Q| \cdot N_\perp)$, where $N_\perp$ is the maximum number of states reachable via the $\perp$ token. Thus, this part takes $O(d \cdot (|Q|^2 + |Q| \cdot |V|))$.

Second, the core dynamic programming update calculates $W[i, q]$ for each diffusion step $i$ and state $q$. This involves iterating over $d$ diffusion steps, $|Q|$ current states $q$, and for each $q$, considering all $|Q|$ possible previous states $q'$. This leads to a complexity of $O(d \cdot |Q|^2)$.

Combining these dominant parts, the total complexity is $O(d \cdot (|Q|^2 + |Q| \cdot |V|) + d \cdot |Q|^2)$, which simplifies to $O(d \cdot (|Q|^2 + |Q| \cdot |V|))$. This can be expressed as $O(d \cdot |Q| \cdot (|Q| + |V|))$.

## 4.5   Context free grammars and reasoning

Currently, DINGO operates with regular grammars. For context-free grammars (CFGs), existing works Ugare et al. [2024b] perform greedy pruning of invalid tokens in a left-to-right manner. The same greedy generation strategy can also be applied to diffusion models, albeit at the cost of optimality. We already have an implementation of greedy left-to-right generation (referred to as Greedy Constrained in Section 5), which naturally extends to CFGs. However, as demonstrated by our experiments, greedy decoding (also employed in more recent works Mündler et al. [2025]) yields significantly worse performance compared to the optimal constrained decoder DINGO for regular expressions. Moreover, DINGO already supports reasoning-augmented grammars Banerjee et al. [2025b], allowing the inclusion of reasoning tokens that do not conform to the output grammar.

# 5 Experiments

In this section, we evaluate DINGO on a math reasoning task (GSM-Symbolic Mirzadeh et al. [2024]) and a schema-based text-to-JSON task (JSONModeEval [NousResearch, 2024]) and demonstrate significant improvement over baselines. In both tasks, we use the LLaDA-8B-Base (LLaDA-8B-B) Nie et al. [2025], LLaDA-8B-Instruct (LLaDA-8B-I) Nie et al. [2025], Dream-v0-Base-7B (Dream-B-7B) Ye et al. [2025], and Dream-v0-Instruct-7B (Dream-I-7B) Ye et al. [2025] models.

**Experimental Setup.** We run experiments on a 48-core Intel Xeon Silver 4214R CPU with 2 Nvidia RTX A5000 GPUs. DINGO is implemented using PyTorch Paszke et al. [2019] and the HuggingFace transformers library Wolf et al. [2020]. The token-level DFA is implemented in Rust using a highly efficient regex-DFA library to minimize overhead during DFA construction and LLM inference. We report the mean number of DFA states and transitions as well as the offline pre-computation time in Appendix E.

**Baselines.** We compare DINGO against unconstrained diffusion LLM generation. Furthermore, to highlight the benefit of optimal constrained decoding with DINGO, we implement a constrained decoding strategy Greedy Constrained that mirrors existing autoregressive constrained generation methods Willard and Louf [2023], Ugare et al. [2024b]. Greedy Constrained iterates over the diffusion block and at each position $i$ computes a binary mask $m \in \{0, 1\}^{|V|}$ based on the DFA, specifying valid tokens ($m = 1$) and excluded tokens ($m = 0$). Decoding is then performed on the masked probability distribution $m \odot v_i$, where $\odot$ denotes element-wise multiplication. Since in some cases, Unconstrained outperforms Greedy Constrained, we also report Best of Greedy + Unconstrained, which takes the better result of the two approaches for each problem in the dataset.

**Math Reasoning:** We evaluate DINGO on GSM-Symbolic Mirzadeh et al. [2024] dataset, which consists of reasoning-based math world problems where numerical values and names are replaced by symbolic variables. Diffusion LLMs are tasked with generating correct symbolic expression solutions to those word problems. We evaluate correctness by using the Z3 solver [De Moura and Bjørner, 2008] to check if the final expressions from the LLM generations are functionally equivalent to the ground truth expressions. We set the generation length to 128, number of blocks to 8, and total diffusion steps to 64 and prompt the LLMs with 4-shot examples from GSM-Symbolic [Mirzadeh et al., 2024] (the prompts can be found in Appendix F.1). We initialize DINGO and Greedy Constrained with a regex (shown in Appendix F.2) that permits math expressions wrapped in « and » and natural language text outside these expressions for reasoning as done in CRANE Banerjee et al. [2025a].

Table 1 compares the performance of DINGO with the baseline methods. The Accuracy (%) column reports the percentage of functionally correct LLM-generated expressions, Parse (%) indicates the percentage of syntactically valid responses (i.e., expressions without invalid operations), and Time provides the average time in seconds taken to generate a completion.

As displayed in the table, DINGO significantly improves functional correctness over the baselines. For instance, for LLaDA-8B-I, DINGO outperforms unconstrained generation by 13 percentage points and Greedy Constrained generation by 5 percentage points. Furthermore, DINGO achieves 100% syntactic accuracy across all models evaluated. On the other hand, unconstrained and Greedy Constrained generation make many syntactic errors, especially for non-instruct tuned models. For these cases, generation with Greedy Constrained results in responses that are syntactically valid prefixes but not syntactically valid by themselves. We present case studies in Appendix F.3. Importantly, DINGO is extremely efficient, introducing marginal overhead compared to unconstrained generation.

**JSON Generation:** We further evaluate DINGO on a text-to-JSON generation task JSON-Mode-Eval, which conists of zero-shot problems specifying a JSON schema and a request to generate a JSON object that contains specified contents. Generating JSON that adheres to a specified schema is extremely important for applications like tool use and function calling Ugare et al. [2024b], Willard and Louf [2023]. We evaluate the correctness of JSON generated by an LLM by first evaluating whether the JSON string can be parsed and converted to a valid JSON object. We further evaluate whether the generated JSON is valid against the schema specified in the prompt. We set the generation length to 128, number of blocks to 1, and the total diffusion steps to 64. For the constrained generation methods, we convert each problem's JSON schema into its corresponding regular expression and guide the diffusion LLM to generate output conforming to that regex.

Table 1: Comparison of constrained and unconstrained generation methods on GSM-Symbolic.

| Model | Method | Acc. (%) | Parse (%) | Time (s) |
|---|---|---|---|---|
| LLaDA-8B-B | Unconstrained | 25 | 54 | 9.06 |
| | Greedy Constrained | 30 | 75 | 9.31 |
| | Best of Greedy + Unconstrained | 30 | 75 | 9.08 |
| | DINGO | **31** | **100** | 9.22 |
| LLaDA-8B-I | Unconstrained | 19 | 35 | 23.78 |
| | Greedy Constrained | 27 | 98 | 23.97 |
| | Best of Greedy + Unconstrained | 27 | 98 | 23.8 |
| | DINGO | **32** | **100** | 23.92 |
| Dream-B-7B | Unconstrained | 17 | 33 | 16.02 |
| | Greedy Constrained | 21 | 41 | 16.13 |
| | Best of Greedy + Unconstrained | 21 | 41 | 16.04 |
| | DINGO | **23** | **100** | 16.19 |
| Dream-I-7B | Unconstrained | 32 | 61 | 23.89 |
| | Greedy Constrained | 34 | 93 | 24.01 |
| | Best of Greedy + Unconstrained | 34 | 93 | 23.9 |
| | DINGO | **36** | **100** | 23.91 |

Table 2 presents the results of our experiment. The Parse (%) column reports the percentage of syntactically valid LLM generations while the Accuracy (%) column reports the percentage of generations that are both syntactically valid and valid against their respective schemas. Notably, DINGO achieves 100% schema validation and syntactic accuracy, while baseline methods struggle in many cases to generate valid JSON. We attribute this to the fact that Greedy Constrained may distort the distribution through its greedy approximation and can only generate a valid prefix, not a fulll parsable generation Park et al. [2024a].

Table 2: Comparison of constrained and unconstrained generation methods for JSON Schema.

| Model | Method | Acc. (%) | Parse (%) | Time (s) |
|---|---|---|---|---|
| LLaDA-8B-B | Unconstrained | 57 | 59 | 6.37 |
| | Greedy Constrained | 80 | 80 | 6.47 |
| | Best of Greedy + Unconstrained | 88 | 90 | 6.41 |
| | DINGO | **100** | **100** | 6.43 |
| LLaDA-8B-I | Unconstrained | 87 | 91 | 6.7 |
| | Greedy Constrained | 78 | 79 | 6.81 |
| | Best of Greedy + Unconstrained | 99 | 99 | 6.73 |
| | DINGO | **100** | **100** | 6.78 |
| Dream-B-7B | Unconstrained | 15 | 18 | 5.31 |
| | Greedy Constrained | 23 | 23 | 5.41 |
| | Best of Greedy + Unconstrained | 32 | 35 | 5.34 |
| | DINGO | **100** | **100** | 5.45 |
| Dream-I-7B | Unconstrained | 85 | 87 | 6.4 |
| | Greedy Constrained | 30 | 30 | 6.51 |
| | Best of Greedy + Unconstrained | 91 | 93 | 6.43 |
| | DINGO | **100** | **100** | 6.55 |

**Comparison with finetuning and rejection sampling:** We compare DINGO with two other possible baselines: (a) fine-tuning on a specific dataset's JSON schema, and (b) rejection sampling, where we iteratively sample responses and check them against a regular expression (referred to as the Sample-Filter approach). For the fine-tuning approach, we perform supervised fine-tuning (SFT) on a dataset of 20k natural language + JSON schema to JSON instance pairs Azinn [2024]. We used the d1 codebase Zhao et al. [2025] for the SFT implementation. While for the rejection sampling approach,

Table 3: Comparison of DINGO with rejection sampling and finetuning.

| Model | Method | Acc. (%) | Parse (%) | Time (s) |
|---|---|---|---|---|
| LLaDA-8B-B | Unconstrained | 57 | 59 | 6.37 |
| | Greedy Constrained | 80 | 80 | 6.47 |
| | Sample-Filter | 68 | 69 | 13.00 |
| | Format Fine-Tuning | 60 | 60 | 6.41 |
| | DINGO | 100 | 100 | 6.43 |
| LLaDA-8B-I | Unconstrained | 87 | 91 | 6.70 |
| | Greedy Constrained | 78 | 79 | 6.81 |
| | Sample-Filter | 93 | 97 | 9.80 |
| | Format Fine-Tuning | 89 | 92 | 6.72 |
| | DINGO | 100 | 100 | 6.78 |

we sampled upto 1000 responses per question. Table 3 shows DINGO significantly outperforms both the baselines.

**Ablation Study on The Number of Diffusion Blocks:** We analyze the performance of DINGO on GSM-Symbolic using different numbers of diffusion blocks. We run generation with a response length of 128, using 64 total diffusion steps, and each of 1, 2, and 8 blocks. As shown in Figure 1, DINGO performs well across all block settings, outperforming baselines in both functional and syntactic correctness. The ablations on the number of diffusion blocks are presented in Appendix I.

**Impact of varying output lengths and different unmasking schedules:** We present an ablation study evaluating different output lengths and three different unmasking strategies: a) random, b) top2 margin, and c) entropy-based unmasking strategies Nie et al. [2025] in Appendix L.

# 6 Related Works

To the best of our knowledge, our work is the first to provide provable guarantees on constrained adherence for inference in diffusion language models. We next discuss the broader set of related works on diffusion language models and constrained language model decoding.

**Diffusion Language Models:** Diffusion Language Models Austin et al. [2021] have emerged as a promising alternative to traditional autoregressive architectures Radford et al. [2019], offering advantages in parallel processing and controllability while addressing limitations in sequential generation. Recent advances in semi-autoregressive diffusion models Han et al. [2023], Nie et al. [2025], Ye et al. [2025], Arriola et al. [2025] have significantly narrowed the performance gap with autoregressive counterparts. SSD-LM [Han et al., 2023] introduced a semi-autoregressive approach that performs diffusion over the natural vocabulary space, enabling flexible output length and improved controllability by iteratively generating blocks of text while facilitating local bidirectional context updates. More recently, several breakthrough models have advanced the field: LLaDA (Large Language Diffusion with mAsking) achieved competitive performance with SOTA open-source autoregressive models of a similar size like LLaMA3-8B through a forward data masking process and

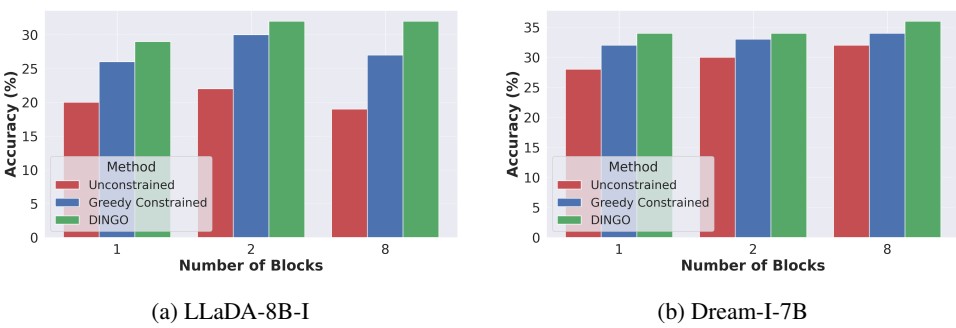

(a) LLaDA-8B-I                    (b) Dream-I-7B

Figure 1: Ablation Study on The Number of Diffusion Blocks For GSM-Symbolic

a reverse process, parameterized by a vanilla Transformer to predict masked tokens [Nie et al., 2025]. BD3-LMs (Block Discrete Denoising Diffusion Language Models)Arriola et al. [2025] introduced a novel approach that interpolates between discrete denoising diffusion and autoregressive models while supporting flexible-length generation and improving inference efficiency with KV caching. Most recently, Dream-7BYe et al. [2025] emerged as a strong open diffusion large language model that matches state-of-the-art autoregressive (AR) language models of similar size.

**Constrained Decoding with Autoregressive LLMs:** Constrained decoding has shown promising results in augmenting autoregressive language models. Researchers have developed efficient techniques for ensuring syntactic correctness in regular [Deutsch et al., 2019, Willard and Louf, 2023, Kuchnik et al., 2023] or context-free [Koo et al., 2024, Ugare et al., 2024a, Dong et al., 2024, Banerjee et al., 2025a] languages. Other works have focused on semantically constrained decoding through Monte Carlo sampling [Lew et al., 2023, Loula et al., 2025] or backtracking [Poesia et al., 2022, Ugare et al., 2025]. Lew et al. [2023], Park et al. [2024a] demonstrated that all these approaches that perform greedy constrained approximation for inference can distort the sampling distribution. DINGO addresses this challenge by performing optimal constrained sampling on blocks of tokens in a diffusion language model, which partially mitigates distribution distortion issues.

Concurrent to our work, Cardei et al. [2025] performs constrained sampling from diffusion language models by minimizing a loss function defined using a surrogate model used for scoring constraints. However, their proposed method does not guarantee convergence to the constraint and necessitates a differentiable surrogate model. In contrast, our work focuses on providing provable guarantees for constraint satisfaction during inference without the need of an additional surrogate model.

**Limitations** DINGO is optimal for per-block generation, making it ideal for pure diffusion settings. However, this optimality may not hold in semi-autoregressive setups involving multiple blocks. Currently, our approach is limited to regular language constraints, while programming languages often belong to context-free or context-sensitive classes. As a result, our method cannot directly enforce these more expressive constraints, which have been addressed in prior work on autoregressive constrained generation. Nonetheless, we believe the core dynamic programming framework behind DINGO can be extended to support richer language classes in future work. Moreover, important constraints like toxicity mitigation fall outside formal language classes, highlighting directions for further research.

# 7 Conclusion

We presented DINGO, a novel dynamic programming approach that enables diffusion LLMs to generate outputs that strictly adhere to regular language constraints while preserving the model's underlying distribution. Our method overcomes the limitations of traditional constrained decoding algorithms that fail with parallel token prediction. Our experimental results on symbolic math and JSON generation tasks demonstrate significant improvements over unconstrained inference, demonstrates that DINGO is an effective solution for structured output generation with diffusion models. Our work bridges an important gap in making diffusion LLMs reliable for applications requiring formal guarantees.

# Acknowledgement

We thank the anonymous reviewers for their insightful comments. This work was supported by funding through NSF Grants No. CCF-2238079, CCF-2316233, CNS-2148583, CCF-2313028, CCF-2217144 and a Research Gift from Amazon AGI Labs.

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

# A  Transformer and Remasking Step

---

**Algorithm 2** Diffusion Step

---

**Require:** Transformer $\mathcal{N}_n$, full output length $n$, prompt $\boldsymbol{p}$, input prompt length $m$, block length $d$, current diffusion step $i$, total diffusion steps $T$, vocabulary $V$.

1: $\boldsymbol{x} \leftarrow \boldsymbol{p} \cdot \perp^d$          $\triangleright$ Pad the input prompt where $n = m + d$
2: $\boldsymbol{v_1} \ldots \boldsymbol{v_n} \leftarrow \mathcal{N}_n(\boldsymbol{x})$          $\triangleright$ $\boldsymbol{v_i} \in \mathbb{R}_+^{|V|}$ output distribution at position $i$
3: $\boldsymbol{l} \leftarrow \text{RemaskPositions}(\boldsymbol{v}_{m+1}, \ldots, \boldsymbol{v}_{m+d}, i, T)$    $\triangleright$ Decides which positions to remask
4: **for** $j \in \boldsymbol{l}$ **do**
5:      $\boldsymbol{v_j} \leftarrow \boldsymbol{0}$
6:      $\boldsymbol{v_j}[\perp] \leftarrow 1$          $\triangleright$ Set probability of all tokens except $\perp$ to 0.
7: $\boldsymbol{r} \leftarrow D_{m,n}(\boldsymbol{v_1} \ldots \boldsymbol{v_n})$   $\triangleright$ Decoding that outputs response with first $m$ tokens are input prompt $\boldsymbol{p}$
8: **return r**

---

We describe the two key components of a single diffusion step: a) **Transformer step:** Computes the output distribution over all tokens in the vocabulary (line 2 Algo. 2). b) **Remasking step:** Based on the output from the transformer step, it greedily decides which token positions to mask. The remasking step can be viewed as updating the output distribution such that, at the masked positions, the mask token $\perp$ is assigned probability 1, while all other tokens receive probability 0 (lines $3 - 6$ Algo. 2). Popular greedy remasking strategies include (line 3 Algo. 2): (i) *Random:* Masks tokens at randomly selected positions Nie et al. [2025]. (ii) *Top token probability:* Masks positions where the top-predicted token has the lowest probability Nie et al. [2025]. (iii) *Entropy-based:* Computes the entropy of the output distribution at each position and masks the positions with the highest entropy Ye et al. [2025].

The number of token positions to remask at the $i$-th step typically depends on the total number of diffusion steps $T$ and the block length $d$. At step 0, all $d$ positions are masked, and the number of masked tokens decreases linearly to 0 over $T$ steps. Thus, at the $i$-th step, the number of masked tokens is given by $\left\lfloor \frac{d \times (T-i)}{T} \right\rfloor$.

# B  Proofs

**Proposition 4.1.** *[Correctness] Given any regular expression $\mathcal{R}$, input prompt $\boldsymbol{p} \in V^m$, block length $d$, output distribution $\mathcal{D}_{m+d} = \boldsymbol{v}_1 \ldots \boldsymbol{v}_{m+d}$, if $L_P(\mathcal{R}) \cap (V \setminus \perp)^d \neq \{\}$ and $\boldsymbol{r} \sim \boldsymbol{v}_{m+1} \ldots \boldsymbol{v}_{m+d}$ be the decoded string, then $\exists \boldsymbol{x} \in V^*.(\boldsymbol{x} \in \mathcal{S}(\boldsymbol{r})) \wedge (\boldsymbol{x} \in L_P(\mathcal{R}))$ holds.*

*Proof.* We assume that $\exists \boldsymbol{x} \in L_P(\mathcal{R}) \cap (V \setminus \perp)^d \wedge (P(\boldsymbol{x}|\boldsymbol{v}_{m+1} \ldots \boldsymbol{v}_{m+d}) \geq 0)$ then the decoded string $\boldsymbol{r}$ satisfy the soundness property (see Definition 3.2). In other words, if there is at least one fully unmasked valid prefix with non-zero probability then DINGO retrieves a valid string.

We show this by induction on the position of tokens. Before moving to the proof, we first define extended transition function $\delta^*$ when $\delta : Q \times V \to 2^Q$ outputs a set of states instead of single state due to mask token $\perp$. In this case, for any string $\boldsymbol{w} \in V^*$, $\delta^*(\boldsymbol{w}, q_0)$ represents the state of reachable states starting from $q_0$. This can be defined as $\delta^*(\{\}, q_0) = \{q_0\}$ and $\delta^*(t_1 \cdots t_{m+1}, q_0) = \cup_{q \in \delta^*(t_1 \cdots t_m, q_0)} \delta(q, t_{m+1})$.

1. Let $0 \leq i \leq d$, and let $t_1 \ldots t_i \in V^i$ denote any token sequence with positive probability mass $\prod_{j=1}^i \boldsymbol{v}_{m+j}[t_j] > 0$. Let $q \in \delta^*(t_1 \cdots t_i, q_0)$. Then, $W[i, q] > 0$. We prove this using induction on $i$.

   (a) Base case $i = 0$: For empty strings only start state $q_0$ is reachable. DINGO initializes $W[0, q_0] = 1 > 0$ and for all $q \neq q_0$, $W[0, q] = 0$. (lines $1 - 3$ in Algo. 1).

   (b) Inductive Step: At position $i + 1$, let $t_1 \ldots t_{i+1} \in V^{i+1}$ s.t. $\prod_{j=1}^{i+1} \boldsymbol{v}_{m+j}[t_j] > 0$. Let $q' \in \delta^*(t_1 \cdots t_i, q_0)$ and $q \in \delta(q', t_{i+1})$. By the inductive hypothesis, for all such $q'$

$W[i, q'] > 0$. Recall,

$$V_{i+1}(q, q') = \begin{cases} \max_{t \in V} \boldsymbol{v}_{m+i+1}(t) \text{ s.t. } q \in \delta(q', t) \\ 0 \text{ if } q, q' \text{ are not connected} \end{cases} \qquad W[i+1, q] = \max_{q' \in Q} W[i, q'] \times V_{i+1}(q, q')$$

Thus, $V_{i+1}(q, q') \geq \boldsymbol{v}_{m+i+1}(t_{i+1}) > 0$ which implies $W[i, q'] \times V_{i+1}(q, q') > 0$. Therefore, $W[i+1, q] = \max_{q' \in Q} W[i, q'] \times V_{i+1}(q, q') > 0$.

2. Since $L_P(\mathcal{R}) \cap (V \setminus \bot)^d \neq \{\}$ by assumption, there exists some $\boldsymbol{y} \in L_P(\mathcal{R}) \cap (V \setminus \bot)^d$. By the Definition 2.6, $q_l = \delta_t^*(\boldsymbol{y}, q_0) \in Q_l$. From the induction above, $W[d, q_l] > 0$. From line 16 in Algo. 1, $q_{max} = \arg\max_{q \in Q_l} W[d, q]$. Thus, by the definition of $\arg\max$, $W[d, q_{max}] \geq W[d, q_l] > 0$.

3. In lines 20-22 in Algo. 1), DINGO reconstructs a d-length sequence $r = t_1 \ldots t_d \in V^d$ such that $q_{max} \in \delta^*(r, q_0)$. For any $t_j \in r$, if $t_j = \bot$, choose any token $\tau_j \in (V \setminus \bot)$ satisfying $\delta_t(q_{j-1}, \tau_j) = q_j$ where $q_j = \delta_t^*(t_1 \ldots t_j, q_0)$. By definition of $\delta_\bot$, $\tau_j$ exists. Substituting every $\bot$ in this manner yields, by Definition 3.1, $\boldsymbol{x} = \boldsymbol{x_1} \ldots \boldsymbol{x_d} \in (V \setminus \bot)^d$. $\boldsymbol{x} \in \mathcal{S}(\boldsymbol{r})$. $\delta_t^*(\boldsymbol{x}, q_0) = q_{max}$. From above, $W[d, q_{max}] > 0$.

4. Since $q_{max} \in Q_l$, by Definition 2.6, $\exists w \in \Sigma^*$ s.t. $\delta^*(\boldsymbol{w}, q_{max}) \in F$. Equivalently, $\boldsymbol{x} \cdot w \in L(\mathcal{R})$, hence $\boldsymbol{x} \in L_P(\mathcal{R})$.

$\square$

**Proposition 4.2.** *[Optimality] Given any regular expression $\mathcal{R}$, input prompt $\boldsymbol{p} \in V^m$, block length $d$, output distribution $\mathcal{D}_{m+d} = \boldsymbol{v}_1 \ldots \boldsymbol{v}_{m+d}$, if $L_P(\mathcal{R}) \cap (V \setminus \bot)^d \neq \{\}$ and $\boldsymbol{r}^* \sim \boldsymbol{v}_{m+1} \ldots \boldsymbol{v}_{m+d}$ be the decoded string, then for any valid string $\boldsymbol{r}'$ satisfying $\exists \boldsymbol{x} \in V^*.(\boldsymbol{x} \in \mathcal{S}(\boldsymbol{r}')) \wedge (\boldsymbol{x} \in L_P(\mathcal{R}))$, $P(\boldsymbol{r}' \mid \boldsymbol{v}_{m+1} \ldots \boldsymbol{v}_n) \leq P(\boldsymbol{r}^* \mid \boldsymbol{v}_{m+1} \ldots \boldsymbol{v}_n)$.*

*Proof.*     1. First, we show that $P(\boldsymbol{r}^* \mid \boldsymbol{v}_{m+1} \ldots \boldsymbol{v}_n) = W[d, q_{max}]$, or equivalently $\prod_{j=1}^d \boldsymbol{v}_{m+j}[\boldsymbol{r_j}^*] = W[d, q_{max}]$. Let $\boldsymbol{r}^* = \boldsymbol{r_1}^* \ldots \boldsymbol{r_d}^*$ and $0 \leq i \leq d$. We prove by induction on $i$ that if DINGO's backtracking (lines $19 - 23$ in Algo 1) has brought us to state $q \in Q$ at position $i$, then $W[i, q] = \prod_{j=1}^i \boldsymbol{v}_{m+j}[\boldsymbol{r_j}^*]$.

   (a) Base case i = 0: $W[0, q_0] = 1 = \prod_{j=1}^0 \boldsymbol{v}_{m+j}[\boldsymbol{r_j}^*]$.
   (b) Inductive Step: At position $i$, let $q', \boldsymbol{r_i}^* = Pr[i, q]$ (line 21 in Algo 1). From lines $14 - 15$ in Algo 1, $W[i, q] = W[i-1, q'] \times \boldsymbol{v}_{m+i}(\boldsymbol{r_i}^*)$. By the inductive hypothesis, $W[i-1, q'] = \prod_{j=1}^{i-1} \boldsymbol{v}_{m+j}[\boldsymbol{r_j}^*]$. Thus, $W[i, q] = \prod_{j=1}^{i-1} \boldsymbol{v}_{m+j}[\boldsymbol{r_j}^*] \times \boldsymbol{v}_{m+i}(\boldsymbol{r_i}^*) = \prod_{j=1}^i \boldsymbol{v}_{m+j}[\boldsymbol{r_j}^*]$.

   Let $q_d \in \delta^*(\boldsymbol{r_1}^* \ldots \boldsymbol{r_d}^*, q_0)$. Since $q_d = q_{max}$ (line 19 in Algo 1), $W[d, q_{max}] = \prod_{j=1}^d \boldsymbol{v}_{m+j}[\boldsymbol{r_j}^*] = P(\boldsymbol{r}^* \mid \boldsymbol{v}_{m+1} \ldots \boldsymbol{v}_n)$.

2. We show that for every valid string $\boldsymbol{r}' = \boldsymbol{r_1}' \ldots \boldsymbol{r_d}'$ satisfying $\exists \boldsymbol{x} \in V^*.(\boldsymbol{x} \in \mathcal{S}(\boldsymbol{r}')) \wedge (\boldsymbol{x} \in L_P(\mathcal{R}))$, $\prod_{j=1}^d \boldsymbol{v}_{m+j}[\boldsymbol{r_j}'] \leq W[d, q_{max}]$. Let $0 \leq i \leq d$ and $q \in \delta^*(\boldsymbol{r_1}' \ldots \boldsymbol{r_i}', q_0)$. We show that $\prod_{j=1}^i \boldsymbol{v}_{m+j}[\boldsymbol{r_j}'] \leq W[d, q]$ using induction on $i$.

   (a) Base case i = 0: $W[0, q_0] = 1 = \prod_{j=1}^0 \boldsymbol{v}_{m+j}[\boldsymbol{r_j}']$.
   (b) Inductive Step: At position $i + 1$, let $q' \in \delta^*(\boldsymbol{r_1}' \cdots \boldsymbol{r_i}', q_0)$ and $q \in \delta(q', \boldsymbol{r_{i+1}}')$. By the inductive hypothesis, $\prod_{j=1}^i \boldsymbol{v}_{m+j}[\boldsymbol{r_j}'] \leq W[i, q']$. Recall,

$$V_{i+1}(q, q') = \begin{cases} \max_{t \in V} \boldsymbol{v}_{m+i+1}(t) \text{ s.t. } q \in \delta(q', t) \\ 0 \text{ if } q, q' \text{ are not connected} \end{cases} \qquad W[i+1, q] = \max_{q' \in Q} W[i, q'] \times V_{i+1}(q, q')$$

   Thus, $\boldsymbol{v}_{m+i+1}(\boldsymbol{r_{i+1}}') \leq V_{i+1}(q, q')$. Hence, $\prod_{j=1}^{i+1} \boldsymbol{v}_{m+j}[\boldsymbol{r_j}'] = \prod_{j=1}^i \boldsymbol{v}_{m+j}[\boldsymbol{r_j}'] \times \boldsymbol{v}_{m+i+1}(\boldsymbol{r_{i+1}}') \leq W[i, q'] \times V_{i+1}(q, q') \leq W[i+1, q]$.

Let $q_d \in \delta^*(\boldsymbol{r_1}' \ldots \boldsymbol{r_d}', q_0)$. Since $\boldsymbol{x} \in V^*.(\boldsymbol{x} \in \mathcal{S}(\boldsymbol{r'})) \wedge (\boldsymbol{x} \in L_P(\mathcal{R}))$, $q_d \in Q_l$. From line 16 in Algo. 1, $q_{max} = \arg\max_{q \in Q_l} W[d, q]$. Thus, by the definition of $\arg\max$, $W[d, q_d] \leq W[d, q_{max}]$. From the inductive hypothesis above, $\prod_{j=1}^d \boldsymbol{v}_{m+j}[\boldsymbol{r_j}'] \leq W[d, q_d] \leq W[d, q_{max}]$.

3. Hence, $P(\boldsymbol{r'} \mid \boldsymbol{v}_{m+1} \ldots \boldsymbol{v}_n) = \prod_{j=1}^d \boldsymbol{v}_{m+j}[\boldsymbol{r_j}'] \leq W[d, q_{max}] = \prod_{j=1}^d \boldsymbol{v}_{m+j}[\boldsymbol{r_j}^*] = P(\boldsymbol{r}^* \mid \boldsymbol{v}_{m+1} \ldots \boldsymbol{v}_n)$.

$\square$

## C   Time complexity analysis of parallelized DINGO DP

---
**Algorithm 3** DINGO DP
---
**Require:** $q_0$, block length $d$, probability vectors $\boldsymbol{v}_1, \ldots \boldsymbol{v}_d$ for the current block, $Q_l, Q, \delta$.
1:  $W[0, q] \leftarrow 0$ for all $(q \in Q) \wedge (q \neq q_0)$
2:  $W[0, q_0] \leftarrow 1$
3:  $Pr[0, q] \leftarrow (\text{None}, \text{None})$ for all $(q \in Q)$  ▷ Initialization of the DP
4:  $V_i \leftarrow \{\}$ for all $i \in \{1, \ldots, d\}$  ▷ maximum token probability transtion $(q' \to q)$ at position $i$
5:  $T_i \leftarrow \{\}$ for all $i \in \{1, \ldots, d\}$  ▷ token for the maximum probability transition $(q' \to q)$
6:  **for** $i \in \{1, \ldots, d\}$ **do**  ▷ The computation along all $d$ can be parallelized
7:      # Parallelize for each $\{1, \ldots d\}$
8:      **for** $(q \in Q)$ **do**
9:          **for** $t \in V$ **do**
10:             $q' \leftarrow \delta(q, t)$
11:             $V_i(q, q'), T_i(q, q') \leftarrow \text{MaxTransition}(\boldsymbol{v}_i, t, q, q')$
12: **for** $i \in \{1, \ldots, d\}$ **do**  ▷ DP computation loop
13:     **for** $(q \in Q) \wedge (q' \in Q)$ **do**
14:         **if** $W[i, q] < W[i - 1, q'] \times V_i(q, q')$ **then**
15:             $W[i, q] \leftarrow W[i - 1, q'] \times V_i(q, q')$  ▷ Update maximum probability path to $q$
16:             $Pr[i, q] \leftarrow (q', T_i(q, q'))$  ▷ Update the parents accordingly
17: $q_{max} \leftarrow \arg\max_{q \in Q_l} W[d, q]$
18: **if** $W[d, q_{max}] = 0$ **then**  ▷ No valid prefixes
19:     **return** None, $q_{max}$
20: $\boldsymbol{r}^* \leftarrow \{\}, q_{curr} \leftarrow q_{max}$
21: **for** $i \in \{d, \ldots, 1\}$ **do**  ▷ Decoding the optimal string $\boldsymbol{r}^*$
22:     $q_{curr}, t \leftarrow Pr[i, q_{curr}]$
23:     $\boldsymbol{r}^* \leftarrow \boldsymbol{r}^* \cdot t$
24: **return** $\text{reverse}(\boldsymbol{r}^*), q_{max}$
---

The parallelism step at line 6 in Algo. 3 can be efficiently implemented using popular frameworks like PyTorch. With parallelism, the computational depth (i.e., the minimum number of sequential steps) reduces to $O\big(\max(|Q|^2, |Q| \times |V|) + |Q|^2 \times d\big)$. For regular expressions, where the number of states $|Q|$ is a small constant, the computational depth becomes $O(|V| + d)$, which is linear in both the vocabulary size $|V|$ and the block length $d$.

## D   Semi-Autoregressive

In the semi-autoregressive setup, given an input $\boldsymbol{p} \in V^m$, the output $\boldsymbol{o} \in V^{m+d \times k}$ is generated over $k$ blocks, where each block is computed via a call to the single block diffusion model. The output of the $i$-th diffusion model call is $\boldsymbol{x}_i = \mathcal{L}_{m_i, n_i}(\boldsymbol{x}_{i-1})$, with $\boldsymbol{x}_0 = \boldsymbol{p}$ and the final output $\boldsymbol{o} = \boldsymbol{x}_k$. The input and output lengths for each block are defined as $m_i = m + (i - 1) \times d$ and $n_i = m + i \times d$ for all $1 \leq i \leq k$.

---

**Algorithm 4** Semi-Autoregressive diffusion LLM Generation

---

**Require:** diffusion LLM $\mathcal{L}$, prompt $\boldsymbol{p}$, answer length $n$, block length $d$, diffusion steps $T$, vocabulary $V$, number of blocks $k$.

1: $\boldsymbol{x} \leftarrow \boldsymbol{p}$            $\triangleright$ Initialize $\boldsymbol{x}$ with input prompt $\boldsymbol{p}$
2: $\boldsymbol{r} \leftarrow \{\}$            $\triangleright$ Intialize the output string
3: **for** $i \in \{1, \ldots, k\}$ **do**
4:      $\boldsymbol{x} \cdot \boldsymbol{r_i} \leftarrow \text{Diffusion}(\boldsymbol{x}, m + (i-1) \times d, d, T, V)$      $\triangleright$ $\boldsymbol{r_i} \in V^d$ is $i$-th output block
5:      $\boldsymbol{r} \leftarrow \boldsymbol{r} \cdot \boldsymbol{r_i}$
6:      $\boldsymbol{x} \leftarrow \boldsymbol{x} \cdot \boldsymbol{r_i}$            $\triangleright$ Compute the input prompt for the next block
7: **Return** $\boldsymbol{r}$

---

---

**Algorithm 5** Semi-Autoregressive Constrained diffusion LLM Generation

---

**Require:** diffusion LLM $\mathcal{L}$, prompt $\boldsymbol{p}$, answer length $n$, block length $d$, diffusion steps $T$, vocabulary $V$, number of blocks $k$, regular expression $\mathcal{R}$.

1: $q_0, Q_l, \delta \leftarrow \text{PreProcess}(\mathcal{R})$      $\triangleright$ Pre-compute the dfa start state, live states and $\delta$
2: $\boldsymbol{x} \leftarrow \boldsymbol{p}$            $\triangleright$ Initialize $\boldsymbol{x}$ with input prompt $\boldsymbol{p}$
3: $\boldsymbol{r} \leftarrow \{\}$            $\triangleright$ Intialize the output string
4: $q_{curr} \leftarrow q_0$            $\triangleright$ Intialize the current dfa state the response is at
5: **for** $i \in \{1, \ldots, k\}$ **do**
6:      $\boldsymbol{x} \cdot \boldsymbol{r_i}, q_{next} \leftarrow \text{Diffusion}(\boldsymbol{x}, m + (i-1) \times d, d, T, V, Q_l, \delta, q_{curr})$
7:      **if** $q_{next} \notin Q_l$ **then**
8:          **return** None            $\triangleright$ No valid completion
9:      $\boldsymbol{r} \leftarrow \boldsymbol{r} \cdot \boldsymbol{r_i}$
10:      $\boldsymbol{x} \leftarrow \boldsymbol{x} \cdot \boldsymbol{r_i}$            $\triangleright$ Compute the input prompt for the next block
11:      $q_{curr} \leftarrow q_{next}$            $\triangleright$ Update current DFA state for next block
12: **Return** $\boldsymbol{r}$

---

In the semi-autoregressive setting, after each block, we ensure that the output generated so far ends in a live state from $Q_l$; otherwise, we return the None string (line 7, Algo. 5). Additionally, we maintain a variable $q_{\text{curr}}$ to track the current DFA state at the end of each block. This state is then used as the starting state for the dynamic programming step in the constrained generation of the next block.

# E Token Transitions Statistics

Table 4: Token Transitions Pre-Computation Statistics

| Model Family | $|V|$ | GSM-Symbolic | | JSON-Mode | |
| --- | --- | --- | --- | --- | --- |
| | | Time(s) | #States | Time(s) | #States |
| LLaDA-8B | 126349 | 32.09 | 40 | 13.22 | 169.31 |
| Dream-7B | 151667 | 37.01 | 40 | 11.87 | 169.31 |

In Table 4, we report the precomputation time and the number of states in the DFA for both tasks. For JSON generation, different regular expressions are used for different schemas; therefore, we report the mean precomputation time and mean number of states. The maximum number of states and precomputation times across all questions are 455 and 17.7 (Dream) 21.3 (LLaDA) seconds, respectively.

# F GSM-Symbolic

## F.1 GSM-Symbolic Prompt

```
You are an expert in solving grade school math tasks. You will be presented
    with a grade-school math word problem with symbolic variables and be
    asked to solve it.

Before answering you should reason about the problem (using the <reasoning>
    field in the response described below). Intermediate symbolic expressions
     generated during reasoning should be wrapped in << >>.

Only output the symbolic expression wrapped in << >> that answers the
    question. The expression must use numbers as well as the variables
    defined in the question. You are only allowed to use the following
    operations: +, -, /, //, %, *, and **.

You will always respond in the format described below:
Let's think step by step. <reasoning> The final answer is <<symbolic
    expression>>

There are {t} trees in the {g}. {g} workers will plant trees in the {g} today
    . After they are done, there will be {tf} trees. How many trees did the {
    g} workers plant today?

Let's think step by step. Initially, there are {t} trees. After planting,
    there are {tf} trees. The number of trees planted is <<tf - t>>. The
    final answer is <<tf - t>>.

If there are {c} cars in the parking lot and {nc} more cars arrive, how many
    cars are in the parking lot?

Let's think step by step. Initially, there are {c} cars. {nc} more cars
    arrive, so the total becomes <<c + nc>>. The final answer is <<c + nc>>.

{p1} had {ch1} {o1} and {p2} had {ch2} {o1}. If they ate {a} {o1}, how many
    pieces do they have left in total?

Let's think step by step. Initially, {p1} had {ch1} {o1}, and {p2} had {ch2}
    {o1}, making a total of <<ch1 + ch2>>. After eating {a} {o1}, the
    remaining total is <<ch1 + ch2 - a>>. The final answer is <<ch1 + ch2 - a
    >>.

{p1} had {l1} {o1}. {p1} gave {g} {o1} to {p2}. How many {o1} does {p1} have
    left?

Let's think step by step. {p1} started with {l1} {o1}. After giving {g} {o1}
    to {p2}, {p1} has <<l1 - g>> {o1} left. The final answer is <<l1 - g>>.

{question}
```

Listing 1: Prompt template for the GSM-Symbolic task Mirzadeh et al. [2024].

## F.2 GSM-Symbolic Regex

```
(?:(?:(?:(?:(?:(?:[  -;=?-~\n]+))*(?:<<(?:(?:\ ))?(?:(?:(?:(?:(?:(?:[a-j])
    |(?:[0-9]{1,3})|\((?:(?:(?:(?:[a-j])|(?:[0-9]{1,3})|\((?:(?:(?:(?:[a-j])
    |(?:[0-9]{1,3})|\((?:(?:(?:(?:[a-j])|(?:[0-9]{1,3})))(?:(?:(?:(?:\ ))
    ?(?:(?:\+|\-|//|/|%|\*|\*\*))(?:(?:\ ))?(?:(?:(?:[a-j])|(?:[0-9]{1,3}))))
    )*)\))))(?:(?:(?:(?:\ ))?(?:(?:\+|\-|//|/|%|\*|\*\*))(?:(?:\ ))?(?:(?:(?:[
    a-j])|(?:[0-9]{1,3})|\((?:(?:(?:(?:[a-j])|(?:[0-9]{1,3})))(?:(?:(?:(?:\ )
    )?(?:(?:\+|\-|//|/|%|\*|\*\*))(?:(?:\ ))?(?:(?:(?:[a-j])|(?:[0-9]{1,3})))
    ))*)\)))))*)\)))(?:(?:(?:(?:\ ))?(?:(?:\+|\-|//|/|%|\*|\*\*))(?:(?:\ ))
    ?(?:(?:(?:[a-j])|(?:[0-9]{1,3})|\((?:(?:(?:(?:[a-j])|(?:[0-9]{1,3})
    |\((?:(?:(?:(?:[a-j])|(?:[0-9]{1,3})))(?:(?:(?:(?:\ ))
    ?(?:(?:\+|\-|//|/|%|\*|\*\*))(?:(?:\ ))?(?:(?:(?:[a-j])|(?:[0-9]{1,3}))))
    )*)\)))(?:(?:(?:(?:\ ))?(?:(?:\+|\-|//|/|%|\*|\*\*))(?:(?:(?:(?:[
    a-j])|(?:[0-9]{1,3})|\((?:(?:(?:(?:[a-j])|(?:[0-9]{1,3})))(?:(?:(?:(?:\ )
    )?(?:(?:\+|\-|//|/|%|\*|\*\*))(?:(?:\ ))?(?:(?:(?:[a-j])|(?:[0-9]{1,3})))
    ))*)\))))))*)\))))))*)\)))(?:(?:(?:(?:\ ))?(?:(?:\+|\-|//|/|%|\*|\*\*))
    (?:(?:\ ))?(?:(?:(?:[a-j])|(?:[0-9]{1,3})|\((?:(?:(?:(?:[a-j])
    |(?:[0-9]{1,3})|\((?:(?:(?:(?:[a-j])|(?:[0-9]{1,3})|\((?:(?:(?:(?:[a-j])
    |(?:[0-9]{1,3})))(?:(?:(?:(?:\ ))?(?:(?:\+|\-|//|/|%|\*|\*\*))(?:(?:\ ))
    ?(?:(?:(?:[a-j])|(?:[0-9]{1,3})))))*)\)))(?:(?:(?:(?:\ ))
    ?(?:(?:\+|\-|//|/|%|\*|\*\*))(?:(?:\ ))?(?:(?:(?:[a-j])|(?:[0-9]{1,3})
    |\((?:(?:(?:(?:[a-j])|(?:[0-9]{1,3})))(?:(?:(?:(?:\ ))
    ?(?:(?:\+|\-|//|/|%|\*|\*\*))(?:(?:\ ))?(?:(?:(?:[a-j])|(?:[0-9]{1,3}))))
    )*)\))))))*)\)))(?:(?:(?:(?:\ ))?(?:(?:\+|\-|//|/|%|\*|\*\*))(?:(?:\ ))
    ?(?:(?:(?:[a-j])|(?:[0-9]{1,3})|\((?:(?:(?:(?:[a-j])|(?:[0-9]{1,3})
    |\((?:(?:(?:(?:[a-j])|(?:[0-9]{1,3})))(?:(?:(?:(?:\ )
    )?(?:(?:\+|\-|//|/|%|\*|\*\*))(?:(?:\ ))?(?:(?:(?:[a-j])|(?:[0-9]{1,3})))
    ))*)\))))))*)\))))))*)\))))))*))(?:(?:\ ))?>>)))+(?:(?:\.))?)
```

Listing 2: GSM-Symbolic Regex

## F.3 GSM-Symbolic Case Studies

**Case Study 1:**

Figure 2: An example from the GSM-symbolic dataset (variables in blue), where unconstrained generation produces syntactically incorrect output, and greedy constrained generation yields a syntactically valid but incorrect answer. In contrast, DINGO generates the correct answer.

**Case Study 2:**

Figure 3: An example from the GSM-symbolic dataset (variables in blue), where unconstrained generation produces syntactically incorrect output, and greedy constrained generation yields a syntactically valid but incorrect answer. In contrast, DINGO generates the correct answer.

# G  JSON-Mode

## G.1  JSON-Mode Example Prompt

```
You are a helpful assistant that answers in JSON. Here's the json schema you
    must adhere to:
<schema>
{'title': 'PromotionalCampaign', 'type': 'object', 'properties': {'campaignID
    ': {'title': 'Campaign ID', 'type': 'string'}, 'productID': {'title': '
    Product ID', 'type': 'string'}, 'startDate': {'title': 'Start Date', '
    type': 'string', 'format': 'date'}, 'endDate': {'title': 'End Date', '
    type': 'string', 'format': 'date'}, 'discountDetails': {'title': '
    Discount Details', 'type': 'string'}}, 'required': ['campaignID', '
    productID', 'startDate', 'endDate']}
</schema>

I'm organizing a promotional campaign for our new eco-friendly laundry
    detergent, which is part of our household products line. The campaign
    will start on June 1, 2023, and end on June 30, 2023. We're planning to
    offer a 15% discount on all purchases during this period. The campaign ID
     is CAMP123456, the product ID is PROD7891011, and the discount details
    are 15% off on all purchases.
Only output the JSON object, no other text or comments.
```

Listing 3: Example JSON Prompt from the JSON-Mode-Eval task NousResearch [2024]. The prompt includes a system message that specifies a schema and a user message that explicitly instructs the model to output a JSON object following that schema with certain parameters.

## G.2  JSON-Mode Example Regex

```
\\{[ ]?"campaignID"[ ]?:[ ]?"([^"\\\\\\x00-\\x1F\\x7F-\\x9F]|\\\\["\\\\])*"[
    ]?,[ ]?"productID"[ ]?:[ ]?"([^"\\\\\\x00-\\x1F\\x7F-\\x9F]|\\\\["\\\\])
    *"[ ]?,[ ]?"startDate"[ ]?:[ ]?"(?:\\d{4})-(?:0[1-9]|1[0-2])
    -(?:0[1-9]|1[-2][0-9]|3[0-1])"[ ]?,[ ]?"endDate"[ ]?:[ ]?"(?:\\d{4})
    -(?:0[1-9]|1[0-2])-(?:0[1-9]|1[-2][0-9]|3[0-1])"([ ]?,[ ]?"
    discountDetails"[ ]?:[ ]?"([^"\\\\\\x00-\\x1F\\x7F-\\x9F]|\\\\["\\\\])*")
    ?[ ]?\\}
```

Listing 4: Regex for the JSON Schema in Appendix  G.2

### G.3 JSON-Mode Case Studies

**Question:** Here's the json schema you must adhere to:<schema> … </schema>. <Question … Please respond with a valid JSON object for this order.> Only output the JSON object, no other text or comments.

**Unconstrained:** {'orderId': 'ORD10234', 'items': [{'name': 'Wireless Mouse', 'price': 15.99}, {'name': 'Bluetooth Keyboard', 'price': 29.99}, {'name': 'Webcam HD', 'price': 45.50}}

Syntax error (missing ']')

**Greedy Constrained:** { "orderId" : "4", 'items': [{'name': 'Wireless Mouse', 'price': 15.99}, {'name': 'Bluetooth Keyboard', 'price': 29.99}, {'name': 'Webcam HD', 'price': 45.501311111111111111111111111111111111111111

Syntax error (incomplete output)

**DINGO:** {'orderId': 'ORD10234', 'items': [{'name': 'Wireless Mouse', 'price': 15.99}, {'name': 'Bluetooth Keyboard', 'price': 29.99}, {'name': 'Webcam HD', 'price': 45.50}]]}

Syntactically Correct

Figure 4: An example from JSON generation, where unconstrained generation produces a syntactically incorrect output, and greedy constrained generation yields a valid but incomplete prefix. In contrast, DINGO generates a syntactically correct answer.

**Question:** Here's the json schema you must adhere to:<schema> … </schema>. Question … service was very prompt and professional, but I wish there was more variety in the product options.'Only output the JSON object, no other text or comments.

**Unconstrained:** \n\n\n"customerID": "CUST29382", "feedbackScore": 4, "comments": "The service was very prompt and professional, but I wish there was more variety in the product options."}

Syntax error (missing '{') 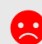

**Greedy Constrained:** { "customerID": "1UST29382", "feedbackScore": 4, \n\n\n\n"comments"\n\n\n\n\n: \n\n\n". Think like the are D to a five year Bible. … a little about the and the human world. Is the author review

Syntax error (incomplete output)

**DINGO:** {'customerID': 'CUST29382', 'feedbackScore': 4, 'comments': 'The service was very prompt and professional, but I wish there was more variety in the product options.'}

Syntactically Correct 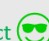

Figure 5: An example from JSON generation, where unconstrained generation produces a syntactically incorrect output, and greedy constrained generation yields a valid but incomplete prefix. In contrast, DINGO generates a syntactically correct answer.

# H   Ablation Study on Number of Blocks for Diffusion LLM Generation (GSM-Symbolic)

We run generation with a response length of 128, using 64 total diffusion steps, and each of 1, 2, and 8 blocks. Table 5 presents the result.

Table 5: Ablation Study on The Number of Diffusion Blocks for GSM-Symbolic

| Model | #Blocks | Method | Acc. (%) | Parse (%) | Time (s) |
|---|---|---|---|---|---|
| LLaDA-8B-I | 1 | Unconstrained | 20 | 54 | 23.66 |
| | | Greedy Constrained | 26 | 94 | 23.7 |
| | | Best of Greedy + Unconstrained | 26 | 94 | 23.66 |
| | | DINGO | 29 | 100 | 23.73 |
| | 2 | Unconstrained | 22 | 54 | 23.63 |
| | | Greedy Constrained | 30 | 96 | 23.81 |
| | | Best of Greedy + Unconstrained | 30 | 96 | 23.65 |
| | | DINGO | 32 | 100 | 23.93 |
| | 8 | Unconstrained | 19 | 35 | 23.78 |
| | | Greedy Constrained | 27 | 98 | 23.97 |
| | | Best of Greedy + Unconstrained | 27 | 98 | 23.8 |
| | | DINGO | **32** | **100** | 23.92 |
| Dream-I-7B | 1 | Unconstrained | 28 | 69 | 23.56 |
| | | Greedy Constrained | 32 | 90 | 23.64 |
| | | Best of Greedy + Unconstrained | 32 | 90 | 23.65 |
| | | DINGO | 34 | 100 | 23.67 |
| | 2 | Unconstrained | 30 | 55 | 23.62 |
| | | Greedy Constrained | 33 | 87 | 23.71 |
| | | Best of Greedy + Unconstrained | 33 | 87 | 23.62 |
| | | DINGO | 34 | 100 | 23.65 |
| | 8 | Unconstrained | 32 | 61 | 23.89 |
| | | Greedy Constrained | 34 | 93 | 24.01 |
| | | Best of Greedy + Unconstrained | 34 | 93 | 23.89 |
| | | DINGO | **36** | **100** | 23.91 |

# I Ablation Study on Number of Blocks for Diffusion LLM Generation (JSON-Mode)

We run generation with a response length of 128, using 64 total diffusion steps, and each of 1, 2, and 8 blocks. Table 6 presents the result.

Table 6: Ablation Study on The Number of Diffusion Blocks for JSON-Mode.

| Model | #Blocks | Method | Acc. (%) | Parse (%) | Time (s) |
|---|---|---|---|---|---|
| LLaDA-8B-I | 1 | Unconstrained | 87 | 91 | 6.7 |
| | | Greedy Constrained | 78 | 79 | 6.81 |
| | | Best of Greedy + Unconstrained | 99 | 99 | 6.73 |
| | | DINGO | 100 | 100 | 6.78 |
| | 2 | Unconstrained | 84 | 92 | 6.72 |
| | | Greedy Constrained | 92 | 94 | 6.83 |
| | | Best of Greedy + Unconstrained | 99 | 99 | 6.73 |
| | | DINGO | 100 | 100 | 6.86 |
| | 8 | Unconstrained | 84 | 89 | 6.73 |
| | | Greedy Constrained | 98 | 98 | 6.87 |
| | | Best of Greedy + Unconstrained | **100** | **100** | 6.75 |
| | | DINGO | 100 | 100 | 6.85 |
| Dream-I-7B | 1 | Unconstrained | 85 | 87 | 6.4 |
| | | Greedy Constrained | 30 | 30 | 6.51 |
| | | Best of Greedy + Unconstrained | 91 | 93 | 6.43 |
| | | DINGO | **100** | **100** | 6.55 |
| | 2 | Unconstrained | 79 | 82 | 6.47 |
| | | Greedy Constrained | 37 | 39 | 6.68 |
| | | Best of Greedy + Unconstrained | 86 | 88 | 6.5 |
| | | DINGO | 100 | 100 | 6.63 |
| | 8 | Unconstrained | 70 | 74 | 6.44 |
| | | Greedy Constrained | 52 | 52 | 6.65 |
| | | Best of Greedy + Unconstrained | 86 | 89 | 6.46 |
| | | DINGO | 100 | 100 | 6.67 |

# J    Ablation Study on Number of Steps for Diffusion LLM Generation (GSM-Symbolic)

We run generation with a response length of 128, 1 block, and each of 16, 32, 64, and 128 total diffusion steps. Table 7 presents the result.

Table 7: Ablation Study on The Number of Diffusion Steps for GSM-Symbolic with Dream-I-7B

| #Steps | Method | Acc. (%) | Parse (%) | Time (s) |
|---|---|---|---|---|
| 16 | Unconstrained | 6 | 20 | 5.99 |
| | Greedy Constrained | 13 | 78 | 6.18 |
| | Best of Greedy + Unconstrained | 13 | 78 | 5.99 |
| | DINGO | **18** | **100** | 6.09 |
| 32 | Unconstrained | 18 | 48 | 11.96 |
| | Greedy Constrained | 25 | 87 | 12.06 |
| | Best of Greedy + Unconstrained | 25 | 87 | 11.96 |
| | DINGO | **28** | **100** | 12.03 |
| 64 | Unconstrained | 28 | 69 | 23.56 |
| | Greedy Constrained | 32 | 90 | 23.64 |
| | Best of Greedy + Unconstrained | 32 | 90 | 23.65 |
| | DINGO | **34** | **100** | 23.67 |
| 128 | Unconstrained | 31 | 74 | 47.83 |
| | Greedy Constrained | 30 | 89 | 47.88 |
| | Best of Greedy + Unconstrained | 31 | 90 | 47.83 |
| | DINGO | **33** | **100** | 47.86 |

# K  Ablation Study on Number of Steps for Diffusion LLM Generation (JSON-Mode)

We run generation with a response length of 128, 1 block, and each of 16, 32, 64, and 128 total diffusion steps. Table 8 presents the result.

Table 8: Ablation Study on The Number of Diffusion Steps for JSON-Mode with Dream-I-7B

| #Steps | Method | Acc. (%) | Parse (%) | Time (s) |
|---|---|---|---|---|
| 16 | Unconstrained | 54 | 59 | 1.51 |
| | Greedy Constrained | 32 | 32 | 1.62 |
| | Best of Greedy + Unconstrained | 68 | 71 | 1.52 |
| | DINGO | **100** | **100** | 1.6 |
| 32 | Unconstrained | 67 | 71 | 3.23 |
| | Greedy Constrained | 35 | 35 | 3.35 |
| | Best of Greedy + Unconstrained | 78 | 82 | 3.24 |
| | DINGO | **100** | **100** | 3.31 |
| 64 | Unconstrained | 85 | 87 | 6.4 |
| | Greedy Constrained | 30 | 30 | 6.51 |
| | Best of Greedy + Unconstrained | 91 | 93 | 6.43 |
| | DINGO | **100** | **100** | 6.55 |
| 128 | Unconstrained | 85 | 87 | 13.42 |
| | Greedy Constrained | 46 | 46 | 13.53 |
| | Best of Greedy + Unconstrained | 95 | 97 | 13.43 |
| | DINGO | **100** | **100** | 13.51 |

# L  Impact of varying output lengths and different unmasking schedules

Table 9: Ablation Study for Different Output Lengths for JSON-Mode. All experiments use a single block, 64 diffusion steps, and generation length of 128.

| Model | Output Length | Method | Acc. (%) | Parse (%) | Time (s) |
|---|---|---|---|---|---|
| LLaDA-8B-I | 128 | Unconstrained | 87 | 91 | 6.70 |
| | 128 | Greedy Constrained | 78 | 79 | 6.81 |
| | 128 | Best of Greedy + Unconstrained | 99 | 99 | 6.73 |
| | 128 | DINGO | 100 | 100 | 6.78 |
| | 256 | Unconstrained | 84 | 87 | 20.25 |
| | 256 | Greedy Constrained | 74 | 76 | 20.62 |
| | 256 | Best of Greedy + Unconstrained | 95 | 96 | 20.29 |
| | 256 | DINGO | 100 | 100 | 20.39 |
| | 512 | Unconstrained | 84 | 84 | 45.77 |
| | 512 | Greedy Constrained | 79 | 80 | 46.24 |
| | 512 | Best of Greedy + Unconstrained | 96 | 96 | 45.83 |
| | 512 | DINGO | 100 | 100 | 46.05 |
| Dream-I-7B | 128 | Unconstrained | 85 | 87 | 6.40 |
| | 128 | Greedy Constrained | 30 | 30 | 6.51 |
| | 128 | Best of Greedy + Unconstrained | 91 | 93 | 6.43 |
| | 128 | DINGO | 100 | 100 | 6.55 |
| | 256 | Unconstrained | 88 | 89 | 17.73 |
| | 256 | Greedy Constrained | 26 | 26 | 18.02 |
| | 256 | Best of Greedy + Unconstrained | 93 | 94 | 17.75 |
| | 256 | DINGO | 100 | 100 | 17.86 |
| | 512 | Unconstrained | 90 | 90 | 36.88 |
| | 512 | Greedy Constrained | 25 | 25 | 37.28 |
| | 512 | Best of Greedy + Unconstrained | 95 | 95 | 36.90 |
| | 512 | DINGO | 100 | 100 | 36.09 |

Table 10: Ablation Study for Different Unmasking Schedules for JSON-Mode.

| Model | Unmasking Type | Method | Acc. (%) | Parse (%) | Time (s) |
|-------|----------------|--------|----------|-----------|----------|
| LLaDA-8B-I | Low-confidence | Unconstrained | 87 | 91 | 6.70 |
| | Low-confidence | Greedy Constrained | 78 | 79 | 6.81 |
| | Low-confidence | Best of Greedy + Unconstrained | 99 | 99 | 6.73 |
| | Low-confidence | DINGO | 100 | 100 | 6.78 |
| | Random | Unconstrained | 80 | 89 | 6.52 |
| | Random | Greedy Constrained | 93 | 95 | 6.74 |
| | Random | Best of Greedy + Unconstrained | 98 | 98 | 6.53 |
| | Random | DINGO | 100 | 100 | 6.76 |
| | Top2 Margin | Unconstrained | 80 | 81 | 7.03 |
| | Top2 Margin | Greedy Constrained | 98 | 98 | 7.17 |
| | Top2 Margin | Best of Greedy + Unconstrained | 100 | 100 | 7.06 |
| | Top2 Margin | DINGO | 100 | 100 | 7.11 |
| | Entropy | Unconstrained | 86 | 89 | 6.89 |
| | Entropy | Greedy Constrained | 71 | 72 | 7.03 |
| | Entropy | Best of Greedy + Unconstrained | 97 | 98 | 6.91 |
| | Entropy | DINGO | 100 | 100 | 7.01 |
| Dream-I-7B | Low-confidence | Unconstrained | 78 | 82 | 6.32 |
| | Low-confidence | Greedy Constrained | 41 | 41 | 6.46 |
| | Low-confidence | Best of Greedy + Unconstrained | 90 | 92 | 6.33 |
| | Low-confidence | DINGO | 100 | 100 | 6.42 |
| | Random | Unconstrained | 55 | 68 | 6.17 |
| | Random | Greedy Constrained | 30 | 30 | 6.28 |
| | Random | Best of Greedy + Unconstrained | 68 | 77 | 6.18 |
| | Random | DINGO | 100 | 100 | 6.30 |
| | Top2 Margin | Unconstrained | 76 | 80 | 6.71 |
| | Top2 Margin | Greedy Constrained | 40 | 40 | 6.85 |
| | Top2 Margin | Best of Greedy + Unconstrained | 89 | 91 | 6.73 |
| | Top2 Margin | DINGO | 100 | 100 | 6.83 |
| | Entropy | Unconstrained | 85 | 87 | 6.40 |
| | Entropy | Greedy Constrained | 30 | 30 | 6.51 |
| | Entropy | Best of Greedy + Unconstrained | 91 | 93 | 6.43 |
| | Entropy | DINGO | 100 | 100 | 6.55 |

