# OpenReview forum: "DINGO: Constrained Inference for Diffusion LLMs"
_NeurIPS.cc/2025/Conference — NeurIPS 2025 poster_

### Official Review · Reviewer_aUe8 · 2025-07-01

**Clarity:** 3
**Significance:** 4
**Originality:** 3
**Rating:** 5
**Confidence:** 4

**Summary:**

This paper proposed an algorithm that guarantees the generated sequence of a diffusion LLM strictly satisfies a given regular expression. The special mask token $\bot$ in diffusion LLM is regarded as a "wildcard token" that can represent any actual token, and the sequence generation task is formulated as a constrained optimization problem to find the sequence with the highest probability while satisfying the constraint of the given regular expression. The regular expression is transformed to a DFA, then a dynamic programming algorithm is proposed to solve the problem. For the $i$-th token position and each state $q$ in the DFA, the algorithm maintains $W[i, q]$, the maximum probability with which a state $q$ can be reached from the start state $q_0$ via transitions on some token sequence with length $i$, which can be computed via the transition function $W[i, q] = \max_{q' \in Q} W[i-1, q'] \cdot V(q', q)$, in which $V(q', q)$ is the maximal transition probability from state $q'$ to state $q$ via a specific token (including via the special mask token $\bot$). Experiments are conducted on JSON and math expressions, with 100% syntactical accuracy and improved functional accuracy.

**Questions:**

No questions yet

**Ethical Concerns:**

["NO or VERY MINOR ethics concerns only"]

**Final Justification:**

I acknowledge that I have read the new experimental result about scalability. I evaluated positively on the paper, and would like to maintain the score.

**Limitations:**

Yes

**Paper Formatting Concerns:**

No concerns

**Quality:**

3

**Strengths And Weaknesses:**

## Strength
- As far as I know, this is the first work to achieve provable constrained decoding on diffusion LLM (another work Cardei et al. [2025] on diffusion LLM encourages but does not guarantee constraint satisfaction)
- The application of dynamic programming improves efficiency
- Efficient implementation with Rust.
- The experimental results look promising, with 100% syntactical accuracy, improved functional accuracy, and no significant increase in time cost.

## Weaknesses
- Only applies to regular language constraints
- Not super scalable

However, these limitations are understandable and have been mentioned in the paper.

---

> ### Author Rebuttal · Authors · 2025-07-31
>
> Thanks for your constructive feedback.
>
> >Q1. Only applies to regular language constraints. Not super scalable. However, these limitations are understandable and have been mentioned in the paper.
>
> **R1:** We have added additional experiments to substantiate the scalability of DINGO with DFAs with larger state sizes.
>
> We randomly generate DFAs with larger state counts, where transitions between states were assigned randomly. As shown in Table 1, DINGO incurs only a marginal additional cost when the generation length scales proportionally with the size of the DFA.
> These experiments were conducted using the same hardware setup as described in the paper. We used a single block, **generation length equal to the number of DFA states**, and the number of diffusion steps set to one-fourth of the generation length. Each experiment was run five times, and we report the 95% confidence interval across these runs.
>
> Table 1: Ablation Study of DINGO overhead for varying number of DFA states
>
> | Model      | # DFA states | Method   | Time (s) |
> |------------|---------|-------------------------------------|----------|
> |LLaDA-8B-I| 10|Unconstrained | 0.08 $\pm$ 0.01 |
> || | Greedy Constrained| 0.08 $\pm$  0.01 |
> || | Diffusion Constrained | 0.08 $\pm$  0.01 |
> || 100| Unconstrained |0.95 $\pm$  0.02 |
> || | Greedy Constrained| 0.98 $\pm$  0.04|
> || | Diffusion Constrained | 1.02 $\pm$  0.04 |
> || 1000|Unconstrained | 57.96 $\pm$  0.32 |
> || | Greedy Constrained| 58.68 $\pm$  0.33 |
> || | Diffusion Constrained |59.33 $\pm$  0.47 |
> || 2000| Unconstrained| 242.96 $\pm$  0.68 |
> || | Greedy Constrained| 244.78 $\pm$  0.91 |
> || | Diffusion Constrained | 245.42 $\pm$  0.75|
> |DREAM-7B-I| 10|Unconstrained | 0.06 $\pm$  0.01 |
> || | Greedy Constrained| 0.06 $\pm$  0.01 |
> || | Diffusion Constrained | 0.06 $\pm$  0.01 |
> || 100| Unconstrained |0.91 $\pm$  0.02 |
> || | Greedy Constrained| 1.02 $\pm$  0.03|
> || | Diffusion Constrained | 0.98 $\pm$  0.02 |
> || 1000|Unconstrained | 62.36 $\pm$  0.51 |
> || | Greedy Constrained| 62.53 $\pm$  0.38|
> || | Diffusion Constrained |62.97 $\pm$  0.27 |
> || 2000| Unconstrained| 264.05 $\pm$  0.84 |
> || | Greedy Constrained| 266.42 $\pm$  0.93 |
> || | Diffusion Constrained | 266.25 $\pm$  0.99 |

---

> ### Comment · Reviewer_aUe8 · 2025-08-05
>
> I acknowledge that I have read the new experimental result about scalability, which has addressed my concern. I evaluated positively on the paper, and would like to maintain the score.

---

> > ### Author Response · Authors · 2025-08-07
> >
> > Thank you for your reply and constructive feedback.

---

### Official Review · Reviewer_tkTe · 2025-07-02

**Clarity:** 4
**Significance:** 2
**Originality:** 3
**Rating:** 5
**Confidence:** 4

**Summary:**

This paper introduces DINGO, a constrained decoding algorithm for diffusion LLMs that supports regular expression. DINGO uses dynamic programming to find optimal output for each masked token while ensuring entire output is a valid prefix within the grammar. Specifically, DP algorithm maintains maximum probability in which a state $q$ can be reached from the initial state $q_0$ via transitions on token sequence of length $i$, and the corresponding transition. The evaluation shows that DINGO effectively eliminates syntactic error in output, without probability distortion happening in constrained decoding for autoregressive models.

**Questions:**

1. Parallelism is a key advantage of diffusion LLMs, enabling time-efficient output generation. It also appears that many parts of the DINGO DP algorithm could also benefit from parallelization (e.g., parallelizing lines 12-15 for the same i). Could you clarify whether the reported number reflect a parallelized implementation, or if they were obtained from sequential computation?

2. DINGO has reasonable overhead for regular expression used in the evaluation. However, the grammar used in the evaluation seems not very complicated and output length seems small. The algorithm’s scalability raises concerns, primarily because most of the computation occurs at inference time. Are there opportunities to save more computation or offload more computation to an offline preprocessing stage?

3. Does 'diffusion step' in lines 233-234 mean 'token index' or 'generation length'? $d$ seems like a output length, not the diffusion steps.

**Ethical Concerns:**

["NO or VERY MINOR ethics concerns only"]

**Final Justification:**

The authors' rebuttal was very helpful in clarifying the implementation of the DINGO algorithm. While DINGO is a straightforward DP-based algorithm for grammatical completion, I believe an efficient Rust implementation and its parallelization are also significant practical contributions. I believe explicitly including these aspects within the paper would further strengthen the paper. Additionally, their responses to other reviewers clarified that generalization to top-k sampling also yields reasonable performance, which effectively addresses a concern I raised in my initial review. I have updated my rating to 5 and look forward to the promised updates.

**Limitations:**

Limitations are adequately addressed.

**Paper Formatting Concerns:**

No major formatting issue was found.

**Quality:**

2

**Strengths And Weaknesses:**

Strengths

- The paper is written clearly.
- The paper proposes first formalization of constrained decoding for diffusion LLMs, with algorithm for regular language.
- The evaluation shows that DINGO has reasonable overhead over unconstrained generation

Weaknesses

- As authors addressed, DINGO is limited to regular language constraints.
- DINGO algorithm only tracks the output with the maximum likelihood, and hence is limited to sampling only the maximum likelihood solution.

---

> ### Author Rebuttal · Authors · 2025-07-31
>
> Thanks for your constructive feedback.
>
> > Q1.  Could you clarify whether the reported number reflect a parallelized implementation, or if they were obtained from sequential computation?
>
> **R1:** Our implementation is highly vectorized and written in TorchScript for better thread-level and kernel-level parallelism [1]. All the reported numbers are from the parallelized implementation, as shown in Appendix C of the supplementary material.
>
> > Q2. Are there opportunities to save more computation or offload more computation to an offline preprocessing stage?
>
> **R2:** We already apply several optimizations offline to reduce runtime overhead. First, we minimize the input DFA, which reduces the number of states while preserving language equivalence. We use the framework from [2] to perform this minimization during the preprocessing stage. Additionally, we cache the DFA's transition table as 1D COO (coordinate list) form tensors [1]. This allows us to exploit sparsity in the state transitions (for instance, eliminating the for loop in line 12 of Algo 1).  There is potential to further improve on these optimizations that can be offloaded to a preprocessing step, and we leave these as future work.
>
>
>
> [1] "PyTorch: an imperative style, high-performance deep learning library", NeurIPS 2019
>
> [2] docs.rs/regex-automata/latest/regex_automata/
>
>
> > Q3. Does 'diffusion step' in lines 233-234 mean 'token index' or 'generation length'? seems like a output length, not the diffusion steps.
>
> **R3:** Thank you for pointing out this typographical error. As you correctly noted, $d$ refers to the length of the generated block. We will fix this in the revised version of the paper.

---

> ### Comment · Reviewer_tkTe · 2025-08-05
>
> The authors' rebuttal was very helpful in clarifying the implementation of the DINGO algorithm. While DINGO is a straightforward DP-based algorithm for grammatical completion, I believe an efficient Rust implementation and its parallelization are also significant practical contributions. I believe explicitly including these aspects within the paper would further strengthen the paper. Additionally, their responses to other reviewers clarified that generalization to top-k sampling also yields reasonable performance, which effectively addresses a concern I raised in my initial review. I have no further questions and have updated my rating to 5, as the rebuttal satisfactorily addressed my concerns.

---

> > ### Author Response · Authors · 2025-08-07
> >
> > Thank you for your reply and constructive feedback. We will incorporate the suggested changes in the revised version of the paper.

---

### Official Review · Reviewer_kh4T · 2025-07-03

**Clarity:** 3
**Significance:** 4
**Originality:** 4
**Rating:** 5
**Confidence:** 3

**Summary:**

DINGO offers an algorithm for regex-constrained inference with diffusion LLMs through dynamic programming.  It is formally correct and optimal.  The empirical results demonstrate gains on benchmarks in math (GSM-Symbolic) and JSON generation.

**Questions:**

1. For multiple blocks, are there cases where you don’t hit 100 parse?  I was looking at Table 4 and expected to see a number below 100, but was surprised it wasn’t there.
2. I didn’t understand the “Best of Greedy + Unconstrained” times – is that only adding up the time from the method chosen per problem?  Part of my confusion was that the ‘best of’ can be lower than Greedy Constrained.
3.  Could you add the computational complexity of the Greedy Constrained method (autoregressive proxy)? Is there an implication for the number of steps used in practice? Greedy Constrained presumably needs to take d steps, so are there situations in which DINGO should be faster than greedy?

**Ethical Concerns:**

["NO or VERY MINOR ethics concerns only"]

**Final Justification:**

My initial positive review of the paper stands, and is only strengthened by the comments from the authors.

**Limitations:**

yes

**Paper Formatting Concerns:**

Neither incredibly major, but just to note:
* Algorithm 2 Diffusion Step from the supplementary material seems like something that belongs in the main paper, but maybe wasn’t included for overflow reasons?  (It’s not referenced in the paper)
* I found the citations difficult to read in-line without some kind of parentheses or coloring to indicate where they are starting

**Quality:**

3

**Strengths And Weaknesses:**

## Strengths
* Novelty & significance: fills a significant current gap (constrained generation for diffusion), and seems like it'll be a baseline/reference for future work in this area.
* Writing/Messaging: States contributions clearly
* Empirical results: Strong results, and included ablations across number of steps, number of blocks, and more.  Very clearly presented.
* Great to have proofs of both correctness and optimality
* Owns limitations of method; demonstrates optimality within constraints.
* Supplementary materials
  * Code is provided.  I didn’t run, but it was clear from the structure and READMEs that thought was put into sharing this with the community and making it easy to reproduce
  * Good case studies demonstrated
  * Stats around states & transitions are nice (supports intuition and suggestions for areas to increase inference speed if willing to sacrifice 100% correctness)

## Weaknesses
* Multiple blocks – showing limitations
  * Empirical results were great, but they didn’t show any deficits
  * It would be great to show extreme samples/constraints where this could be suboptimal
* Be more clear about limitations of regex-based constraints - this was mentioned at the end of the paper, but would be great to cover in the intro more thoroughly to motivate and constrain the work.

## Small suggestions
* Define DFA before using the acronym; would be helpful to define “live state” in more colloquial terms as well (for readers from LLM rather than formal logic backgrounds)
* Little thing: “Algorithm 1 Dingo DP” should probably have the acronym spelled out for readers who are skimming your paper on the first pass
* Related works just before the conclusion was a strange choice to me – seems like it belongs before the background for this paper?
* Demonstrate the flexibility and limitations of regex-based constraints (as opposed to other methods) through a few example samples that aren't derived from benchmarks
* [nit] "fulll parsable generation" (full)

---

> ### Author Rebuttal · Authors · 2025-07-31
>
> Thanks for your constructive feedback.
>
> > Q1. Extreme samples/constraints where this could be suboptimal
>
> **R1:** In the practical cases considered in the paper, we did not observe any parsing errors across multiple blocks. However, theoretically, parsing errors can occur when the generation is **only** a valid prefix of a parsable string but not parsable on its own. A simple example is when the total generation length is shorter than the length of the shortest acceptable string in the regular language. In such cases, DINGO can only generate a valid prefix of a parsable string, but not a complete one.
>
> > Q2. Be more clear about limitations of regex-based constraints.
>
> **R2:** Thank you for the suggestion. We will include a more detailed discussion of the limitations of regex-based constraints in the revised version of the paper.
>
> >Q3. For multiple blocks, are there cases where you don’t hit 100 parse?
>
> **R3:** Please refer to the response to Q1.
>
> > Q4. "Best of Greedy + Unconstrained" times – is that only adding up the time from the method chosen per problem? Part of my confusion was that the ‘best of’ can be lower than Greedy Constrained.
>
> **R4:** In the "Best of Greedy + Unconstrained" approach, we run both the Greedy and Unconstrained methods in parallel and report the runtime of the method that completes first and returns a parsable string. For this reason, the runtime of the "Best of" approach can be lower than that of Greedy Constrained. We will clarify this in the revised version of the paper.
>
>
> > Q5. Could you add the computational complexity of the Greedy Constrained method (autoregressive proxy)?
>
> **R5:** Assuming that the token mask indicating the set of syntactically invalid tokens at each position is precomputed \[1], the complexity of greedy constrained decoding is $O(d \times |V|)$, where $d$ is the block size and $|V|$ is the vocabulary size. While the greedy approach is more efficient, it is not optimal and, as demonstrated by our experiments, yields suboptimal results.
>
>
> [1] SynCode: LLM Generation with Grammar Augmentation, TMLR, 2025.
>
> > Q6. Other Suggestions.
>
> **R6:** Thank you for pointing this out. We will address these issues in the revised version of the paper.

---

> > ### Comment · Reviewer_kh4T · 2025-08-07
> >
> > Thank you for addressing my comments.
> >
> > I think it'd be great to add the clarification on R1 in the paper as well.
> >
> > Given my initial positive review of the paper, and after reviewing the author's comments to other reviewers' concerns, I'm satisfied as long as the clarifications are added to the paper.

---

> > > ### Author Response · Authors · 2025-08-07
> > >
> > > Thank you for your reply and constructive feedback. We will incorporate the suggested changes in the revised version of the paper.

---

### Official Review · Reviewer_TRQv · 2025-07-03

**Clarity:** 3
**Significance:** 3
**Originality:** 3
**Rating:** 5
**Confidence:** 4

**Summary:**

This paper tries to address a critical limitation of diffusion-based Large Language Models (LLMs): their inability to adhere to formal syntactic constraints during their parallel token generation process. The authors propose DINGO, a constrained decoding algorithm specifically designed for diffusion LLMs. DINGO utilizes a dynamic programming approach to find the single most probable output string that strictly conforms to a user-specified regular expression. The method is provably correct (always producing a valid string) and optimal (finding the string with the highest probability under the model's distribution). The authors demonstrate DINGO's effectiveness on challenging structured generation tasks like symbolic math and JSON generation, where it achieves significant accuracy improvements over unconstrained decoding with minimal computational overhead.

**Questions:**

I know that there are a few works trying to do constrained decoding in an asymptotically unbiased way. Can you take a look and see if similar things can be applied here? [Efficient and Asymptotically Unbiased Constrained Decoding for Large Language Models]

**Ethical Concerns:**

["NO or VERY MINOR ethics concerns only"]

**Final Justification:**

The authors have addressed most of my concerns, and, conditioned on the fact that this response can be incorporated into the revised paper, I think it is a good work for the conference.

**Limitations:**

Yes, the authors have discussed limitations and I agree with the authors.

**Paper Formatting Concerns:**

NaN

**Quality:**

3

**Strengths And Weaknesses:**

### Strengths

- **Novel and Important Problem Formulation:** The paper tackles a novel and important problem. As diffusion LLMs become more prominent due to their efficiency, the lack of methods to control their output structure is a major barrier to their adoption in high-stakes applications. DINGO is the first work to formally address constrained decoding in this non-autoregressive setting.
- **Provable Guarantees:** A significant strength of this work is its theoretical rigor. Unlike heuristic-based approaches, DINGO comes with formal proofs for both correctness and optimality . It guarantees that the output will always be valid and will be the single most likely valid sequence according to the model's predictions, which is a very strong claim.
- **High Empirical Effectiveness:** The experimental results are compelling. The substantial performance gains—up to 68 percentage points on challenging JSON generation tasks—clearly demonstrate the practical necessity and effectiveness of the proposed method. Achieving 100% syntactic and schema validity where unconstrained models fail highlights the real-world value of DINGO .
- **Efficient Implementation:** The dynamic programming approach is computationally efficient, with a complexity polynomial in the block length and the number of DFA states . The experiments confirm that this adds only marginal overhead compared to unconstrained inference, making it a practical solution

### Weaknesses

- First, the algorithm's optimality guarantee leads to a **bias in the output distribution**. DINGO deterministically returns the single most probable valid sequence . While this is ideal for tasks where one best answer is desired, it is a significant modification for a generative model. A truly unbiased constrained generation process would sample from the model's original distribution conditioned on the valid set (i.e., sampling from the renormalized distribution over all valid strings). DINGO instead collapses this rich distribution to a single point (its mode). This limits its use in applications requiring diversity or stochasticity, and it would be beneficial to discuss the potential of extending the framework to support true unbiased sampling.
- Second, the algorithm's complexity, while polynomial, **scales with the number of states in the constraint's DFA**. This means that while DINGO is efficient for constraints describable by small DFAs, its practicality may diminish for very complex regular expressions that compile to a large number of states. This limitation is even more pronounced for more expressive language classes like Context-Free Grammars (CFGs), which the paper acknowledges. Since a CFG parser can have a potentially infinite state space (when considering the stack), the current DP approach is not directly applicable, limiting its utility for tasks like generating code in many programming languages.
- **Last is about insufficiency of baselines**. While I understand that as a novel algorithm, there might not be strong existing algorithms to perform against, I do think there is a few necessary evaluations. For instance, sample-then-filter and format fine-tuning is very standard and should be considered. Training-based methods can serve as a ceiling baselines, while comparing with training-free methods are necessary. I think the authors should add a few straightforward approaches in order to demonstrate the cost with DP.

---

> ### Author Rebuttal · Authors · 2025-07-31
>
> Thanks for your constructive feedback.
>
> >Q1. Diverse topk Sampling with DINGO.
>
> **R1:** Although the current dynamic programming (DP) approach is designed to sample the syntactically valid string with the highest probability, it is possible to extend it to draw the top-k syntactically valid strings according to the predicted output distributions. At a high level, in the modified algorithm, for each position and DFA state, we maintain the top-k valid paths from the start state, instead of only the highest-probability path as in the current implementation. For subsequent positions, the top-k paths are iteratively constructed using the top-k paths of all states at the current position and the predicted token distribution at the next position. The cost of top-k sampling grows linearly with respect to k (i.e., k times the complexity of top-1 sampling). We will include the formal proof of correctness and the time complexity of the top-k sampling algorithm in the revised version of the paper.
>
> We have implemented top-k using the same hardware setup described in the paper. Table 1 summarizes the results on the GSM-Symbolic dataset. DINGO outperforms the unconstrained approach in both Pass\@k and parse rate (percentage of samples that are syntactically valid out of all drawn samples). Note that the parse rate decreases as $k$ increases, since the unconstrained approach tends to sample more syntactically invalid responses with larger $k$, whereas DINGO maintains its accuracy.
>
> Table 1: Ablation Study For Top-K Sampling with DINGO and Unconstrained on GSM-Symbolic. Parse Rate represents % of all samples that are syntactically correct, respectively, for a given problem averaged over all problems.
>
> | Model      | k | Method                              | Pass@k (%) | Parse Rate (%) | Time (s) |
> |------------|---------|-------------------------------------|----------|-----------|----------|
> |   LLaDA-8B-B         | 10      | Unconstrained                       |  25      |  43.7       |   10.6   |
> |            |         | DINGO                               |  31    |   100     |  11.32    |
> |          | 50      | Unconstrained                       |      25  |   34.6    |   10.69  |
> |            |         | DINGO                               |  31    |    100    |  11.44    |
> | LLaDA-8B-I | 10      | Unconstrained                       |   19   |    38.2     |   23.14   |
> |           |         | DINGO                               |  32   |   100     |   23.68   |
> |  | 50      | Unconstrained                       |            19      |  28.38     |  23.18    |
> |            |         | DINGO                               |    32   |  100      |    23.83  |
>
>
> > Q2. Scalability with large states in the DFA.
>
> **R2:** We already report the number of states in the DFAs used in our experiments, with the largest DFA containing 455 states.
> To further evaluate the scalability of DINGO, we randomly generated DFAs with larger state counts, where transitions between states were assigned randomly. As shown in Table 2, DINGO incurs only a marginal additional cost when the generation length scales proportionally with the size of the DFA.
> These experiments were conducted using the same hardware setup as described in the paper. We used a single block, **generation length equal to the number of DFA states**, and the number of diffusion steps set to one-fourth of the generation length. Each experiment was run five times, and we report the 95% confidence interval across these runs.
>
> Table 2: Ablation Study of DINGO overhead for varying number of DFA states
>
> | Model      | # DFA states | Method   | Time (s) |
> |------------|---------|-------------------------------------|----------|
> |LLaDA-8B-I| 10|Unconstrained | 0.08 $\pm$ 0.01 |
> || | Greedy Constrained| 0.08 $\pm$  0.01 |
> || | Diffusion Constrained | 0.08 $\pm$  0.01 |
> || 100| Unconstrained |0.95 $\pm$  0.02 |
> || | Greedy Constrained| 0.98 $\pm$  0.04|
> || | Diffusion Constrained | 1.02 $\pm$  0.04 |
> || 1000|Unconstrained | 57.96 $\pm$  0.32 |
> || | Greedy Constrained| 58.68 $\pm$  0.33 |
> || | Diffusion Constrained |59.33 $\pm$  0.47 |
> || 2000| Unconstrained| 242.96 $\pm$  0.68 |
> || | Greedy Constrained| 244.78 $\pm$  0.91 |
> || | Diffusion Constrained | 245.42 $\pm$  0.75|
> |DREAM-7B-I| 10|Unconstrained | 0.06 $\pm$  0.01 |
> || | Greedy Constrained| 0.06 $\pm$  0.01 |
> || | Diffusion Constrained | 0.06 $\pm$  0.01 |
> || 100| Unconstrained |0.91 $\pm$  0.02 |
> || | Greedy Constrained| 1.02 $\pm$  0.03|
> || | Diffusion Constrained | 0.98 $\pm$  0.02 |
> || 1000|Unconstrained | 62.36 $\pm$  0.51 |
> || | Greedy Constrained| 62.53 $\pm$  0.38|
> || | Diffusion Constrained |62.97 $\pm$  0.27 |
> || 2000| Unconstrained| 264.05 $\pm$  0.84 |
> || | Greedy Constrained| 266.42 $\pm$  0.93 |
> || | Diffusion Constrained | 266.25 $\pm$  0.99 |
>
>
> > Q3. Comparison with other potential baselines
>
> **R3:** We have added additional experiments demonstrating the effectiveness of DINGO compared to other potential approaches, including fine-tuning and rejection sampling (referred to as **Sample-Filter**). In all cases, DINGO significantly outperformed the baselines (Table 3). We have used the same hardware setup as described in the paper.
>
>
> For the **Sample-Filter** approach, we iteratively sample a response and use a DFA to check if the sampled response is a syntactically valid prefix. If not, we reject the respons and sample the next one. We sample up to 1000 responses.
>
> For the **Fine-Tuning** approach, we perform supervised fine-tuning (SFT) on a dataset of 20k natural language + JSON schema to JSON instance pairs [1]. We used the d1 codebase [2] for the SFT implementation.
>
> For all experiments, we used a single block, a generation length of 128, and 64 diffusion steps.
>
>
> Table 3: Comparison of Different Methods for JSON-Mode.
>
> | Model      | Method         | Acc. (%) | Parse (%) | Time (s) |
> |------------|----------------|----------|-----------|----------|
> | LLaDA-8B-B | Unconstrained  | 57       | 59        | 6.37     |
> | | Greedy Constrained  | 80       | 80        | 6.47     |
> | | Sample-Filter  |    68    |   69      |  13.00    |
> | | Format Fine-Tuning  |    60    |     60    |  6.41   |
> | | DINGO  | 100       | 100        | 6.43     |
> | LLaDA-8B-I | Unconstrained  | 87       | 91        | 6.70     |
> | | Greedy Constrained  | 78       | 79        | 6.81     |
> | | Sample-Filter  |     93   |   97      |  9.8   |
> | | Format Fine-Tuning  |    89    |    92     |  6.72    |
> | | DINGO  | 100       | 100        | 6.78     |
>
> [1] huggingface.co/datasets/ChristianAzinn/json-training
>
> [2] github.com/dllm-reasoning/d1/tree/main
>
>
> > Q4. I know that there are a few works trying to do constrained decoding in an asymptotically unbiased way. Can you take a look and see if similar things can be applied here? [Reference: Efficient and Asymptotically Unbiased Constrained Decoding for Large Language Models]
>
> **R4:** Thank you for the reference. However, the proposed approach provides only asymptotic guarantees, whereas DINGO is guaranteed to always draw the highest (or top-k) probability string(s) that forms a valid prefix of a parsable string in the language. That said, DINGO currently supports only regular expressions. For more expressive syntactic constraints such as context-free grammars (CFGs), the referenced method could serve as a promising direction for future work.

---

> > ### Comment · Reviewer_TRQv · 2025-08-05
> >
> > I thank the authors for their response. My concerns are mostly addressed. The Scalability with the number of states in DFA should be theoretically and empirically analyzed in the revised paper.

---

> > > ### Author Response · Authors · 2025-08-07
> > >
> > > Thank you for your reply and constructive feedback. We will incorporate the suggested changes in the revised version of the paper.

---

### Official Review · Reviewer_t6qZ · 2025-07-03

**Clarity:** 3
**Significance:** 3
**Originality:** 3
**Rating:** 5
**Confidence:** 2

**Summary:**

This paper proposes DINGO, a distribution-preserving, provably correct constrained decoding algorithm for diffusion LLMs. It compiles user-specified regular expressions into DFAs and applies a Viterbi-style dynamic programming step at each unmasking to enforce constraints without altering the model’s distribution. Experiments demonstrate substantial gains in output validity with only modest extra computation.

**Questions:**

What is the computational complexity when multiple constraints must be enforced simultaneously, and how does it scale with the number or size of DFAs?

**Ethical Concerns:**

["NO or VERY MINOR ethics concerns only"]

**Final Justification:**

I have read the author's rebuttal and the comments from other reviewers, and after a comprehensive evaluation, I have decided to maintain my original score.

**Limitations:**

Yes

**Quality:**

3

**Strengths And Weaknesses:**

Strengths
1. Seamlessly integrates DFA constraints with dynamic programming into parallel diffusion decoding, providing formal guarantees of both validity and distribution fidelity.
2. Demonstrates substantial validity improvements on realistic tasks, with only a small increase in runtime relative to native sampling.

Weaknesses
1. The paper does not include a thorough empirical evaluation against other constrained decoding approaches for diffusion models.
2. The impact of varying output lengths and different unmasking schedules on constraint enforcement is not fully explored.

---

> ### Author Rebuttal · Authors · 2025-07-31
>
> Thanks for your constructive feedback.
>
> > Q1. Empirical evaluation against other constrained decoding approaches for diffusion models.
>
> **R1:** DINGO is the first constrained decoding algorithm for diffusion LLMs. However, we have added additional experiments demonstrating the effectiveness of DINGO compared to other potential approaches, including fine-tuning and rejection sampling (referred to as **Sample-Filter**). In all cases, DINGO significantly outperformed the baselines (Table 1). We have used the same hardware setup as described in the paper.
>
> For the **Sample-Filter** approach, we iteratively sample a response and use a DFA to check if the sampled response is a syntactically valid prefix. If not, we reject the response and sample the next one. We sample up to 1000 responses.
>
> For the **Fine-Tuning** approach, we perform supervised fine-tuning (SFT) on a dataset of 20k natural language + JSON schema to JSON instance pairs [1]. We used the d1 codebase [2] for the SFT implementation.
>
> For all experiments, we used a single block, a generation length of 128, and 64 diffusion steps.
>
> Table 1: Comparison of Different Methods for JSON-Mode.
> | Model      | Method         | Acc. (%) | Parse (%) | Time (s) |
> |------------|----------------|----------|-----------|----------|
> | LLaDA-8B-B | Unconstrained  | 57       | 59        | 6.37     |
> | | Greedy Constrained  | 80       | 80        | 6.47     |
> | | Sample-Filter  |    68    |   69      |  13.00    |
> | | Format Fine-Tuning  |    60    |     60    |  6.41   |
> | | DINGO  | 100       | 100        | 6.43     |
> | LLaDA-8B-I | Unconstrained  | 87       | 91        | 6.70     |
> | | Greedy Constrained  | 78       | 79        | 6.81     |
> | | Sample-Filter  |     93   |   97      |  9.8   |
> | | Format Fine-Tuning  |    89    |    92     |  6.72    |
> | | DINGO  | 100       | 100        | 6.78     |
>
> [1] huggingface.co/datasets/ChristianAzinn/json-training
>
> [2] github.com/dllm-reasoning/d1/tree/main
>
> > **Q2. Impact of varying output lengths and different unmasking schedules.**
>
> **R2:** We present an ablation study evaluating different output lengths and unmasking strategies. Using the same experimental setup as in the paper, DINGO consistently outperforms the baselines across all configurations (Tables 2 and 3). We have used the same hardware setup as described in the paper. All experiments in Table 2 were conducted with a single block, and the number of diffusion steps was always set to half the generation length.
>
> Table 2: Ablation Study For Different Output Lengths for JSON-Mode.
>
> | Model      | Output Length | Method                              | Acc. (%) | Parse (%) | Time (s) |
> |------------|---------|-------------------------------------|----------|-----------|----------|
> | LLaDA-8B-I  | 128      | Unconstrained                       | 87       | 91        | 6.70     |
> |            |         | Greedy Constrained                  | 78       | 79        | 6.81     |
> |            |         | Best of Greedy + Unconstrained      | 99       | 99        | 6.73     |
> |            |         | DINGO                               | 100      | 100       | 6.78     |
> |            | 256       | Unconstrained                       | 84       | 87        | 20.25     |
> |            |         | Greedy Constrained                  | 74       | 76        | 20.62     |
> |            |         | Best of Greedy + Unconstrained  | 95  | 96  | 20.29     |
> |            |         | DINGO                               | 100      | 100       | 20.39     |
> |            | 512      | Unconstrained                       | 84       | 84        | 45.77     |
> |            |         | Greedy Constrained                  | 79       | 80        | 46.24     |
> |            |         | Best of Greedy + Unconstrained      | 96       | 96        | 45.83    |
> |            |         | DINGO                               | 100      | 100       | 46.05     |
> |   Dream-I-7B         | 128      | Unconstrained                       | 85       | 87        | 6.4     |
> |            |         | Greedy Constrained                  | 30       | 30        | 6.51     |
> |            |         | Best of Greedy + Unconstrained      | 91       | 93        | 6.43     |
> |            |         | DINGO                               | 100      | 100       | 6.55     |
> |            | 256       | Unconstrained                       | 88       | 89        | 17.73     |
> |            |         | Greedy Constrained                  | 26       | 26        | 18.02     |
> |            |         | Best of Greedy + Unconstrained      | 93       | 94        | 17.75    |
> |            |         | DINGO                               | 100      | 100       | 17.86     |
> |  | 512       | Unconstrained                       | 90       | 90        | 36.88     |
> |            |         | Greedy Constrained                  | 25       | 25        | 37.28     |
> |            |         | Best of Greedy + Unconstrained      | 95       | 95        | 36.9     |
> |            |         | DINGO                               | 100      | 100       | 36.09    |
>
> All experiments in Table 2 use a single block, 64 diffusion steps and generation length of 128.
>
> Table 3: Ablation Study For Different Unmasking Schedules for JSON-Mode.
>
> | Model      | Unmasking Type | Method                              | Acc. (%) | Parse (%) | Time (s) |
> |------------|---------|-------------------------------------|----------|-----------|----------|
> | LLaDA-8B-I | Low-confidence      | Unconstrained                       | 87       | 91        | 6.70     |
> |            |         | Greedy Constrained                  | 78       | 79        | 6.81     |
> |            |         | Best of Greedy + Unconstrained      | 99       | 99        | 6.73     |
> |            |         | DINGO                               | 100      | 100       | 6.78     |
> |            | Random       | Unconstrained                       | 80       | 89        | 6.52     |
> |            |         | Greedy Constrained                  | 93       | 95        | 6.74     |
> |            |         | Best of Greedy + Unconstrained      | 98       | 98        | 6.53     |
> |            |         | DINGO                               | 100      | 100       | 6.76     |
> |            | Top2 Margin       | Unconstrained                       | 80       | 81        | 7.03     |
> |            |         | Greedy Constrained                  | 98       | 98        | 7.17     |
> |            |         | Best of Greedy + Unconstrained  | 100 | 100   | 7.06     |
> |            |         | DINGO                               | 100      | 100       | 7.11     |
> |            | Entropy       | Unconstrained                       | 86       | 89        | 6.89     |
> |            |         | Greedy Constrained                  | 71       | 72        | 7.03     |
> |            |         | Best of Greedy + Unconstrained      | 97       | 98        | 6.91     |
> |            |         | DINGO                               | 100      | 100       | 7.01     |
> |   Dream-I-7B         | Low-confidence       | Unconstrained                       | 78       | 82        | 6.32     |
> |            |         | Greedy Constrained                  | 41       | 41        | 6.46     |
> |            |         | Best of Greedy + Unconstrained      | 90       | 92        | 6.33     |
> |            |         | DINGO                               | 100      | 100       | 6.42     |
> |            | Random       | Unconstrained                       | 55       | 68        | 6.17     |
> |            |         | Greedy Constrained                  | 30       | 30        | 6.28     |
> |            |         | Best of Greedy + Unconstrained      | 68       | 77        | 6.18     |
> |            |         | DINGO                               | 100      | 100       | 6.30     |
> |            | Top2 Margin       | Unconstrained                       | 76       | 80        | 6.71     |
> |            |         | Greedy Constrained                  | 40       | 40        | 6.85     |
> |            |         | Best of Greedy + Unconstrained      | 89       | 91        | 6.73     |
> |            |         | DINGO                               | 100      | 100       | 6.83     |
> |  | Entropy       | Unconstrained                       | 85       | 87        | 6.40     |
> |            |         | Greedy Constrained                  | 30       | 30        | 6.51     |
> |            |         | Best of Greedy + Unconstrained      | 91       | 93        | 6.43     |
> |            |         | DINGO                               | 100      | 100       | 6.55     |
>
> > Q3. What is the computational complexity when multiple constraints must be enforced simultaneously, and how does it scale with the number or size of DFAs?
>
> **R3:** Currently, DINGO supports a single regular expression. Theoretically, it can support multiple regular expressions enforced simultaneously, since regular languages are closed under intersection \[1], and multiple regular constraints can be represented by a single DFA. However, with a naive implementation, the number of states $Q_A$ in the resulting DFA $A$, obtained by intersecting DFAs $A_1$ and $A_2$ with state sets $Q_{A_1}$ and $Q_{A_2}$ , can be as large as $|Q_A| = |Q_{A_1}| \times |Q_{A_2}|$ in the worst case \[1]. This is indeed a tight bound, and state explosion cannot be avoided in the worst case \[1]. Note that in practice, the DFA $A$ typically is not minimal, and a much smaller equivalent DFA can often be obtained using DFA minimization techniques [2]. We already use the framework from [2] to minimize the input DFA in the preprocessing step.
> We believe that efficiently supporting multiple regular constraints while maintaining both optimality and correctness is an interesting direction for future work.
>
> [1]"On the State Complexity of Intersection of Regular Languages", ACM SIGACT News, 1991. \
> [2] docs.rs/regex-automata/latest/regex_automata/

---

> > ### Comment · Reviewer_t6qZ · 2025-08-05
> >
> > Thanks for the detailed response. My concern has been addressed. As my original score was already positive, I will maintain it.

---

> > > ### Author Response · Authors · 2025-08-07
> > >
> > > Thanks for your reply and constructive feedback.

---

### Note · Authors · 2025-08-13

We thank the ACs and reviewers for their thoughtful and constructive feedback. We are encouraged that they recognized the novelty, significance, provable guarantees, and strong empirical performance of our work. We also note that after our rebuttal, all reviewers indicated that their concerns had been addressed. Below, we summarize our responses to the main points raised:

**1. Empirical evaluation against additional baselines**

In the rebuttal, we added experiments comparing DINGO to format fine-tuning and rejection sampling, in addition to the unconstrained generation and autoregressive constrained decoding proxy results already in the paper. In all cases, DINGO significantly outperforms these baselines.

**2. Impact of output length and unmasking schedules**

We conducted further evaluations across different output lengths and unmasking schedules. In every configuration, DINGO consistently outperforms the baselines.

**3. Diverse top‑k sampling with DINGO**

We extended DINGO to generate the top‑k syntactically valid strings based on the predicted output distributions. The modified algorithm scales linearly with k in computational cost. In the revised paper, we will include a formal proof of correctness and the time complexity analysis for top‑k sampling. Our implementation outperforms unconstrained generation across all tested k values in both pass@k and parse rate.

**4. Scalability of DINGO**

As reported in the paper, the largest DFA in our experiments contains 455 states. To further assess scalability, we generated random DFAs with larger state counts and randomized transitions. We observed only marginal additional computational cost when sequence length scales proportionally with DFA size. We also detail offline optimizations to reduce runtime overhead, including DFA state minimization and storing the DFA transition table in 1D COO (coordinate list) tensor form to exploit transition sparsity. We further clarify that the runtime numbers reported in the paper reflect a parallelized implementation.

---

### Decision · Program_Chairs · 2025-09-17

**Decision:**

Accept (poster)

**Comment:**

The paper studies constrained decoding for diffusion models, which is an interesting and timely topic to study. The proposed algorithm is based on the conditional independency property of diffusion language models and uses dynamic programming to efficiently enforce constraints while offers strong theoretical guarantee in terms of the final decoded sequence. The author provides theoretical justifications on the algorithm's correctness and optimality. Experiments on math reasoning and text-to-json generation tasks are provided to demonstrate their advantage over unconstrained decoding and baselines base on greedy resampling.

The reviewers all appreciated the novelty of the paper, its solid theoretical guarantee, and strong empirical performance. During the discussion phase, concerns were raised about lack of comparison to reasonable baselines, the limitation of the method to sample the maximum likelihood sequence, and scalability of the method to large state size. The reviewers were satisfied after the authors provided experimental comparison with other baseline algorithms including sample-filter and format fine-tuning, presented an extension to sequence-level top-k sampling, and discussed its scalability implications.

Overall, there is a consensus among the reviewers that the paper makes a solid contribution and should be highlighted in the conference. Hence I would recommend an acceptance for the paper. The authors should include the provided additional experiments and incorporate the reviewers' suggestions in the final revision.